

# Leaf Area Index Changes Explain GPP Variation across an Amazon Drought Stress Gradient

Sophie Flack-Prain[1], Patrick Meir[1,2], Yadvinder Malhi[4], Thomas Luke Smallman[1,3], Mathew Williams[1,3]

[1] School of GeoSciences, University of Edinburgh, Edinburgh, UK

[2] Research School of Biology, Australian National University, Canberra, ACT, Australia

[3] National Centre for Earth Observation, University of Edinburgh, UK

[4] Environmental Change Institute, School of Geography and the Environment, University of Oxford, Oxford, UK

*Correspondence to:* Sophie Flack-Prain (s.flack-prain@ed.ac.uk)



## Abstract

The capacity of Amazon forests to sequester carbon is threatened by climate change-induced shifts in precipitation patterns. However, the relative importance of plant physiology, ecosystem structure, and trait composition responses in determining variation in GPP, remain largely unquantified, and vary among models. We evaluate the relative importance of key climate constraints to gross primary productivity (GPP), comparing direct plant physiological responses to water availability and indirect

structural and trait responses (via changes to leaf area index (LAI), roots and photosynthetic capacity). To separate these factors we combined the Soil-Plant-Atmosphere model with forcing and observational data from seven intensively studied forest plots along an Amazon drought stress gradient. We also used machine learning to evaluate the relative importance of individual climate factors across sites. Our model experiments showed that variation in LAI was the principal driver of differences in GPP across

the gradient, accounting for 33% of observed variation. Differences in photosynthetic capacity ($V_{cmax}$ and $J_{max}$) accounted for 21% of variance, and climate (which included physiological responses) accounted for 16%. Sensitivity to differences in climate was highest where shallow rooting depth was coupled with high LAI. On sub-annual timescales, the relative importance of LAI in driving GPP increased with drought stress ($R^2=0.72$), whilst the importance of solar radiation decreased ($R^2=0.90$).

Given the role of LAI in driving GPP across Amazon forests, improved mapping of canopy dynamics is critical, opportunities for which are offered by new satellite-based remote sensing missions such as GEDI, Sentinel and FLEX.

    Keywords: Canopy Dynamics, Leaf Traits, Tropical Rainforests, Precipitation, Gross Primary

Productivity



## 1. Introduction

As the entry point for carbon into the biosphere, gross primary productivity (GPP) is central to the global carbon cycle. Tropical rainforests alone account for one third of total terrestrial photosynthesis assimilating ~41 Pg of carbon each year (Beer et al., 2010). Carbon fluxes across the tropics are tightly coupled to climate, and water availability is a principal driver of spatial and temporal variation in photosynthesis (Fisher et al., 2007, Von Randow et al., 2013, Beer et al., 2010, Malhi et al., 2015, Guan et al., 2015). Across Amazon forests, GPP decreases linearly with increasing seasonal water deficit (Malhi et al., 2015). Shifts in precipitation patterns as a result of anthropogenic climate change are predicted to have a major impact on Amazon GPP (Phillips et al., 2009, Malhi et al., 2008, Meir and Woodward, 2010, Zhang et al., 2015, Meir et al., 2015a). Longer and more intense dry seasons are projected, together with an increased frequency and severity of drought events (Joetzjer et al., 2013, Boisier et al., 2015, Duffy et al., 2015). Given the biogeochemical influence of Amazon forests at regional and global scales (Liu et al., 2017), accurately predicting GPP response to drought stress is critical.

Dynamic global vegetation models (DGVMs) disagree on the effects of projected precipitation change on Amazon carbon dynamics. Galbraith et al. (2010) found future shifts in precipitation patterns had little effect on model estimates of biomass change (for two of the three models tested), reflecting poorly the observed sensitivity of Amazon forests to water availability illustrated by through-fall exclusion experiments and natural drought events (Rowland et al., 2015a, Nepstad et al., 2007, Phillips et al., 2009). Substantial progress has been made in model development to capture the impact of drought stress on plant physiology. By coupling stomatal conductance and plant hydraulic theory, models have proved better able to predict ecosystem functioning and mortality (Eller et al., 2018, Fisher et al., 2018, Fisher et al., 2006, Fisher et al., 2007, Bonan et al., 2014). However, the interactions between drought stress, ecosystem structure (e.g. canopy dynamics and rooting depth) and trait composition (e.g. $V_{cmax}$, $J_{max}$, leaf lifespan and leaf mass per unit area (LMA)), are typically absent from models, despite having a major impact on simulated GPP (Fauset et al., 2012, Sakschewski et al., 2016, Lee et al., 2013).



Furthermore, changes in canopy dynamics have been identified as a likely cause for the disparity between field observations and model predictions (Restrepo-Coupe et al., 2017, Powell et al., 2013).

The relative importance of plant physiology, ecosystem structure, and trait composition responses in determining variation in GPP, remain largely unquantified in data-constrained analysis (Meir et al., 2015b). Plant physiological responses to drought stress include stomatal conductance, which is limited by water availability and atmospheric demand. Stomatal conductance constrains GPP via changes in $CO_2$ supply, but is considered a short (varying on sub-hourly timescales), rather than long-term response to climate forcings (Sperry et al., 2002). Changes to ecosystem structure and traits, such as LAI, rooting depth and carboxylation capacity, are expected to be more longstanding (Meir et al., 2015a).

Extensive evidence links spatial and temporal variation in drought stress with ecosystem structure. Leaf area index (LAI) typically decreases with increasing drought stress (Iio et al., 2014, Meir et al., 2015b, Brando et al., 2008, Grier and Running, 1977, Wright et al., 2013). LAI determines the surface area for GPP, impacting light capture capacity. Near surface root mass, length and surface area decline with seasonal drought stress (and increase during periods of high soil water availability to exploit available resources), whilst deep roots can support water supply during dry periods (Nepstad et al., 1994, Metcalfe et al., 2008). Root depth, mass and traits influence hydraulic supply and consequently stomatal conductance.

Leaf traits similarly exhibit spatial and temporal variation with changing water availability. Leaf nitrogen content (per unit mass), light- and $CO_2$-saturated photosynthetic rates (per unit mass) increase with drought stress across tropical precipitation gradients, whilst p50 (the water potential at which 50% of hydraulic conductivity is lost) declines (Wright et al., 2004, Santiago et al., 2004, Anderegg, 2015). Leaf traits affect GPP via photosynthetic capacity ($V_{cmax}$ and $J_{max}$) (Bahar et al., 2017, Fyllas et al., 2017), and through their influence on canopy carbon economics, via leaf growth and maintenance costs (Bloom et al., 1985). However, understanding the interactions between photosynthetic drivers across different spatial and temporal scales is limited.



Field observations show variation in Amazon GPP is correlated with physiological, ecosystem structure and trait composition responses to climate (Restrepo-Coupe et al., 2013, Goulden et al., 2004, Hutyra et al., 2007, Wu et al., 2017, Wagner et al., 2017). Modelling approaches have similarly highlighted the role of canopy dynamics and leaf traits in driving spatial and temporal variation in GPP (Mercado et

al., 2011, Castanho et al., 2013, Restrepo-Coupe et al., 2013, Rodig et al., 2018), however their relative effects have not been explicitly isolated and quantified. Quantifying the direct effect of discrete photosynthetic drivers has been limited by the need for detailed field measurements of carbon fluxes, canopy dynamics and traits. Furthermore, whilst a deserved research effort has focused on the importance of nutrient availability in driving spatial variation in GPP (Mercado et al., 2011, Castanho

et al., 2013), the role of ecosystem responses to water availability has received less attention (Green et al., 2019). In light of projected changes in rainfall patterns across the basin, capturing responses to water availability in ecosystem models is critical to reducing current uncertainty around Amazon climate-vegetation feedbacks. We aim to reduce the uncertainty by assessing the relative effects of physiological, structural and trait responses to water availability on GPP across monthly to annual

timescales.

We apply a validated ecosystem model to plots across the Amazon, spanning a large drought stress gradient, and a range in forest types from moist equatorial to seasonally dry tropical forests. Process modelling allows the links between climate, ecosystem structure and leaf traits to be quantified explicitly, and separated, across timescales (Figure 1). The soil plant atmosphere model (SPA)

(Williams et al., 1996, Williams et al., 1998, Fisher et al., 2006, Fisher et al., 2007, Rowland et al., 2015b) is well suited to this investigation given its prior use in accurately simulating carbon and water fluxes in Amazon tropical forests. We link the modelling to data gathered over multiple years (2009-2010) on permanent sample plots from the Global Ecosystems Monitoring (GEM) network (Doughty et al., 2015a, Malhi et al., 2015). The datasets comprise detailed measurements of carbon fluxes, carbon

stocks and leaf traits, and were used to constrain the SPA model. We simulate the effect of forest structure and leaf trait distributions along the drought stress gradient, and explore the covariation of



observed leaf traits (leaf N content (a proxy for photosynthetic capacity) and LMA) and those derived from model calibrations (leaf lifespan), before using SPA to address the following questions:

1. Is spatial variation in GPP across the drought stress gradient principally driven by the direct effects of climate and soils, which include physiological responses to water availability via hydraulic transport and stomatal conductance? Alternatively, are indirect effects of climate, via structural and trait responses to water availability (LAI, rooting biomass, root depth and photosynthetic capacity i.e $V_{cmax}$ and $J_{max}$), more important?

2. Does the sensitivity of GPP to differences in climate, LAI, photosynthetic capacity ($V_{cmax}$ and $J_{max}$) and rooting depth vary across the drought stress gradient?

3. What drives seasonal variation in GPP across an Amazon forest drought stress gradient?

Linked to question one, we hypothesise that indirect effects of climate via structural and trait responses are more important than the direct effects (via physiological responses), in explaining spatial variation in GPP across the drought stress gradient (Figure 1). We further posit that LAI is the principal driver of differences in GPP among Amazon forests, effected through the observed increase in leaf area with decreasing drought stress.

For question two, we predict that the sensitivity of GPP to differences in climate, LAI, photosynthetic capacity ($V_{cmax}$ and $J_{max}$) and rooting depth will vary dependent on water demand (via LAI and stomatal conductance) and supply (climate and root depth and biomass; Figure 1). We expect that forests under lower drought stress will be most sensitive to differences in LAI and photosynthetic capacity within the bounds of observations across the gradient. We predict that forests under higher drought stress will be more sensitive to differences in rooting depth. We expect forests with high LAI but shallow rooting depth will be most sensitive to differences in climate, due to their higher transpiration demand but low capacity for water supply.

For question three, we hypothesise that on monthly timescales, climate will be more important than canopy dynamics in driving GPP. Across the drought stress gradient, we expect that solar radiation will be relatively more important during the wet season, whilst VPD will be more important during the dry season, reflecting seasonal shifts in light and water availability. Due to differences in dry season length,



we predict that for forests experiencing lower drought stress, solar radiation will be most important in driving sub-annual variation in GPP, whilst for forests under higher drought stress, VPD will be the dominant driver.

By combining detailed plot-level timeseries data with a hydrodynamic terrestrial ecosystem model, we are able to use an innovative model experimentation approach to understand the drivers of spatial variation in GPP, beyond correlative effects. We are able to apportion variation in GPP to the direct and indirect effects of climate (Figure 1), across sub-annual and annual timescales (Q1 and Q3). Furthermore, by performing a sensitivity analysis within the context of observed variation in parameters across the Amazon (Q2) we identify areas potentially more vulnerable to changes in precipitation regime.

## 2. Methods

We parameterised and validated the ecosystem model SPA to permanent sample plots along an Amazon mean Maximum Climatological Water Deficit (MCWD) gradient (-85 to -498 mm yr$^{-1}$) for the years 2009-2010. Plot characteristics are summarised in Table 1, and detailed in full in the supplementary material. MCWD is a measure of peak seasonal water deficit, where more negative MCWD indicate higher drought stress. We used MCWD instead of annual precipitation as water deficit is more closely linked to mechanisms constraining GPP than total water input. We analysed the distribution of LAI and leaf traits across the MCWD gradient. We then undertook a series of model experiments to: (i) apportion spatial variation in GPP to drivers (climate, soils, LAI, rooting biomass and depth and photosynthetic capacity); (ii) investigate how the sensitivity of GPP to differences in drivers varies across the MCWD gradient; and (iii) quantify the importance of LAI, VPD, solar radiation, precipitation and air temperature in driving sub-annual variation in GPP using the random forest machine learning technique (Breiman, 2001).

### 2.1 The Soil Plant Atmosphere model (SPA)

The Soil-Plant-Atmosphere model (SPA) is a hydrodynamic terrestrial ecosystem model, which has been calibrated and evaluated for moist tropical forests in Manaus and Caxiuanã (Williams et al., 1996, Williams et al., 1998, Fisher et al., 2007). In SPA, carbon and water fluxes are estimated through



process-based modelling of radiative transfer, boundary layer and stomatal conductance, plant and leaf ecophysiology and soil-plant energy and water balance (Smallman et al., 2013, Williams et al., 1996). Plant physiological responses to water availability are well represented in SPA due to the stomatal conductance algorithm being coupled directly to plant water use (Fisher et al., 2006). Within SPA, C

allocation between structural tissue and the non-structural C (NSC) pool is executed via the sub model DALEC$_{canopy}$ (Bloom and Williams, 2015) (Figure 2). Leaf maintenance respiration was calculated as a function of leaf N content (Reich et al., 2008) and total leaf C stock (see supplementary material). Within SPA, wood and fine root maintenance respiration were simulated as a function of component C stock and a plot specific respiration coefficient. Growth respiration was calculated as fixed fraction of

NPP (0.28) (Waring and Schlesinger, 1985).

## 2.2 Model Calibration

Following data collation to parameterise SPA, the model was calibrated and validated for each plot prior to conducting model experiments. Measurements used to parameterise SPA include: soil texture, soil C stock, leaf N content, LMA, photosynthetic capacity, the fraction of NPP allocated to fine roots

and wood, root depth, and foliar, wood and fine root C stocks (Table 2). Soil, wood and fine root C stocks (single point measurements, not timeseries) were initial model inputs and allowed to vary thereafter dependent on simulated C dynamics. Plot specific field measurements of leaf N content are presented in Fyllas et al. (2009), or where absent were retrieved from trait databases using plot species composition (Kattge et al., 2011; Poorter and Bongers, 2006). Photosynthetic capacity estimates ($V_{cmax}$

and $J_{max}$) were derived from leaf N content (Walker et al., 2014), or field measurements (Caxiuanã only). Wood and root respiration measurements were used together with component C stocks to estimate plot specific wood and root respiration coefficients.

The model was driven using hourly meteorological data, retrieved from local weather stations. Short gaps in air temperature, wind speed, shortwave radiation and vapour pressure deficit measurements (<6

hours), were filled by spline interpolation between existing data. Where local meteorological data was unavailable for a longer period of time, or for gaps in precipitation measurements, hourly spline-interpolated ERA-Interim data were used (Dee et al., 2011). The interpolation of solar radiation





estimates accounted for the solar zenith angle. MCWD was calculated for the years 2009-2010 as the minimum monthly water deficit reached within the year, where monthly water deficit is equal to the

previous month's water deficit, plus precipitation, minus evapotranspiration (Aragao et al., 2007). Calculated MCWD was consistent with previously published estimates for all plots excluding Caxiuanã, which were calculated across different years (Malhi et al., (2015), Caxiuanã -203mm, Tambopata -259mm, Kenia -386mm, Tanguro -482mm; this study, Caxiuanã -85±65mm, Tambopata -265±59mm, Kenia 342±146mm, Tanguro 451±73mm).

The simulation of soil water drainage in SPA was calibrated against timeseries of field measurements of soil moisture. Initial investigations comparing modelled soil moisture to monthly field data highlighted an overestimation by SPA. The empirical model used in SPA to relate soil texture to water retention (Saxton et al., 1986, eqn. 10) was then calibrated by adjusting the slope of the interaction to better represent soil moisture across tropical soils (to within standard error estimates of mean annual

soil moisture).

Leaf litterfall parameters (day of peak leaf fall, leaf fall period and leaf lifespan) were calibrated against field data to accurately simulate litterfall period and amplitude (within standard error estimates of annual litterfall). Wood and fine root biomass turnover rates were estimated assuming each forest ecosystem was at steady state given the maturity of stands and their disturbance history:

$$turnover\ rate_i \propto \frac{NPP_i}{C\ stock_i}$$

Where $i$ is wood or fine roots.

Local LAI estimates derived from hemispherical photographs were used to force simulated LAI. Leaf NPP was calculated as the difference between the foliar C stock of the previous timestep and that which would equate to field measured LAI. Leaf NPP was allocated prior to other plant components, and if

the leaf NPP requirement exceeded total NPP for the given timestep, the non-structural C pool was drawn upon (where total NPP was calculated as the difference between simulated GPP and autotrophic respiration) (see supplementary material). Under the assumption that allocation to NSC is an active



process and that the pool serves functions additional to the seasonal redistribution of C (Dietze et al., 2014), depletions in the NSC pool induce redirection of a fraction of NPP towards NSC storage to maintain a stable NSC pool. Root and wood NPP were calculated from the NPP remaining after leaf allocation.

## 2.3 Model Validation

For each plot, SPA calibrations were constrained by the upper and lower sample error of LAI measurements to produce an estimate of model uncertainty. However, given we do not quantify intrinsic model error beyond that associated with parameter estimates, we recognise that the model error estimates presented are underestimated. Observation constrained SPA simulations were then validated against biometric field measurements of C fluxes (i.e. from infra-red gas analysers, dendrometers, root ingrowth cores litterfall traps etc.). Linear regression models were constructed to compare modelled estimates and independent field measurements of GPP, autotrophic respiration and total NPP. A comprehensive comparison of model estimates and independent field measurements of component NPP and respiration were also made. Validation of the SPA model against biometric data lent confidence to subsequent analyses, where the model was used to explore C fluxes under non-observed conditions.

## 2.4 Model Experiments

Our aim was to isolate the direct effects of climate and soils (via physiological responses), ecosystem structure, and leaf traits on simulated GPP. To avoid capturing the feedback effects of changing photosynthate supply (i.e. as a result of changes in climate, soils, ecosystem structure or traits) on ecosystem structure, model experiments were conducted in the absence of C cycle feedbacks. Thus, within model experiments, C stocks for each component (leaves, wood, fine root, coarse root) were constrained to observations unless otherwise stated.

### 2.4.1 Experiment 1. Drivers of Spatial Variation in GPP

Through a series of model input alternations, we used SPA to quantify the effects of (i) climate, (ii) soil properties, (iii) LAI, (iv) root biomass and (v) rooting depth, and (vi) trait responses driven by photosynthetic capacity ($V_{cmax}$ and $J_{max}$), on simulated GPP. Model inputs for each driver were alternated at each plot, to that of all other plots, and annual GPP values for each of the two years





retrieved. For example, plot CAX04 was simulated with the climate, soil properties, LAI, root biomass, root depth and photosynthetic capacity of CAX06, TAM05, TAM06, KEN01, KEN02, and Tanguro etc. SPA simulated GPP for a total of 462 combinations (for climate, 7 plots × 3 alternations × 2 years, plus for the remaining drivers, 5 drivers × 7 plots × 6 alternations × 2 years) were combined with 14 annual GPP estimates from observation constrained (control) runs (7 plots × 2 years). A factorial

ANOVA was applied to the difference between GPP from each model run and its control simulation (n=476, i.e. 462 +14) (Galbraith et al., 2010). The proportions of variation in GPP explained by climate, soil properties, LAI, photosynthetic capacity, root biomass and rooting depth, were then calculated as the conditional sum of square divided by the total sum of squares.

### 2.4.2 Experiment 2. Variation in Forest Sensitivity to Drivers of GPP

We quantified how the relative sensitivity of GPP to differences in LAI, climate, photosynthetic capacity and rooting depth varied across the MCWD gradient. For example, we tested whether forests occupying lower drought stress zones were more sensitive to differences in LAI than forests in higher drought stress zones, etc. We used model outputs generated in *Experiment 1* to calculate the sensitivity of GPP to drivers at each plot, within the bounds of observations across the MCWD gradient. Root

biomass and soil properties were not included in the analysis as across the MCWD gradient they explained little variation in GPP (*Experiment 1*, Table 6). The sensitivity of GPP to drivers at each plot was calculated as the absolute range in simulated GPP values under each driver alternation i.e. the sensitivity of CAX04 to variation in LAI was calculated as the maximum GPP minus the minimum GPP simulated by alternating LAI to that of all other plots etc.  Plots were grouped by location

(Caxiuanã, Tambopata, Kenia and Tanguro) to compare how the sensitivity of GPP to LAI, climate, photosynthetic capacity and rooting depth varies across the MCWD gradient.

### 2.4.3 Experiment 3. Drivers of Sub-Annual Variation in GPP

We quantified the role of climate and LAI in explaining variation in sub-annual GPP. We used the random forest technique to compute the relative importance of LAI, VPD, solar radiation, precipitation

and air temperature driving variation in monthly GPP (n=168; 7 plots × 24 months), where GPP estimates were derived from SPA simulations. To quantify the effects of LAI and climate variables on

monthly GPP we used the random forest machine learning technique applied by means of the Python
Scikit-Learn module (Breiman, 2001, Pedregosa et al., 2011). The approach uses multiple mathematical
decision tree predictors to describe a dependent variable as a function of selected independent variables.

An importance value between 0 and 100 was assigned to each driver based on a tree-wise comparison
of explanatory power (Moore et al., 2018, López-Blanco et al., 2017). We calculated the average relative
importance of drivers at each plot to determine the principal drivers of variation in sub-annual GPP and
investigated the seasonality of driver importance.

## 3. Results

Following model calibration, validation and an investigation into the distribution of LAI and leaf traits
across the MCWD gradient, we (i) quantify the drivers of spatial variation in GPP, (ii) compute the
variation in forest sensitivity to drivers of GPP, and (iii) calculate the relative importance of drivers of
sub-annual variation in GPP.

### 3.1 Model Calibration

Calibrated SPA soil water content corresponded well to field measurements from the GEM network
(Figure 3). Simulated mean annual soil moisture estimates were within field measurement standard
error for all plots.The timing of observed peak soil moisture was captured by SPA simulations ($R^2$=0.98,
p<0.001, RMSE=1 month). SPA simulated seasonal soil moisture range exhibited a non-significant,
positive correlation with field measurements ($R^2$=0.35, p=0.21, RMSE=5%). Notably, for some plots

such as Kenia, the magnitudes of seasonal peak soil water fluxes were not captured by SPA simulations
(up to 39% lower than field estimates), whilst for Tanguro, peak soil water lasted 3 months longer in
SPA simulations than was measured in the field.

SPA was also successfully calibrated to simulate local leaf litterfall accurately. The calibration of leaf
fall cycle parameters in SPA using GEM leaf litterfall timeseries (Table 4), resulted in the magnitude
and timing of leaf litterfall being well represented by the model for all plots (monthly leaf litterfall range

for GEM measurements and SPA simulations $R^2$=0.54, p=0.009, RMSE= 11.2 gC m$^{-2}$ yr$^{-1}$; timing of



leaf litterfall peak $R^2$=0.96, p<0.001, RMSE=1.1 months) (Figure 4). SPA-simulated mean annual leaf litterfall correlated significantly with GEM estimates ($R^2$=0.99, p=<0.001, RMSE=9.0 gC m$^{-2}$ yr$^{-1}$).

## 3.2 Model Validation

Estimates of ecosystem-scale C fluxes from SPA model runs were validated against biometrically derived estimates from the GEM network. GPP$_{SPA}$ and GPP$_{GEM}$ estimates were correlated across plots, though not significantly ($R^2$=0.36, p=0.15; Figure 5a). Along the MCWD gradient, GPP$_{SPA}$ estimates varied across plots by 1137 gC m$^{-2}$ yr$^{-1}$, whilst GPP$_{GEM}$ estimates varied by 1202 gC m$^{-2}$ yr$^{-1}$. Error bars overlap between GPP$_{SPA}$ and GPP$_{GEM}$ estimates for all plots except KEN01 and TAM06, though

marginally (difference KEN01 115 gC m$^{-2}$ yr$^{-1}$, TAM06 50 gC m$^{-2}$ yr$^{-1}$). GPP$_{GEM}$ error bars are field estimate standard error, and GPP$_{SPA}$ error bars represent simulated GPP variance under LAI standard error. Across plots, GPP$_{SPA}$ estimates were 0.57% higher than GPP$_{GEM}$ estimates. The correlation between GPP and MCWD was similar for GPP$_{SPA}$ ($R^2$=0.64, p=0.03, slope=2.4) and GPP$_{GEM}$ estimates ($R^2$=0.52, p=0.07, slope=2.00).

NPP$_{SPA}$ estimates (the sum of model simulated root and wood NPP and data-constrained leaf NPP) were also correlated with NPP$_{GEM}$ measurements across plots ($R^2$=0.38, p=0.14), though not significantly due to differences in Kenia plots (on exclusion of Kenia plots $R^2$=0.92, p= 0.01, RMSE=42 gC m$^{-2}$ yr$^{-1}$) (Figure 5b). NPP$_{SPA}$ estimates were 7.9% lower than field measurements across plots on average. Ra$_{SPA}$ (the sum of predicted leaf respiration, and parameterised root and wood respiration) were significantly

correlated with biometric measurements (Ra$_{GEM}$) across plots ($R^2$=0.59, p=0.04; Figure 5c), though were on average 5.3% higher.

Leaf respiration estimates simulated as a function of leaf nitrogen content were correlated with field measurements, though not significantly ($R^2$=0.47, p=0.09; Table 5). Parameterised wood and fine root respiration, together with fine root NPP, correlated significantly with field measurements. SPA

estimates of wood NPP did not correlate significantly with GEM measurements due to underestimation at KEN01 (on exclusion $R^2$=0.78, p=0.02, RMSE=7.5 gC m$^{-2}$ yr$^{-1}$). Further comparisons of SPA estimates and GEM measurements of component NPP and respiration are presented in Table 5.



### 3.4 LAI and Leaf Traits Trends along the MCWD gradient

Field estimated mean annual LAI ranged from 2.2 to 5.2 $m^2$ $m^{-2}$, and increased (though not significantly)

with MCWD across plots ($R^2$=0.35, p=0.16). A negative, non-significant correlation existed between calibrated leaf lifespan and MCWD ($R^2$=0.50, p=0.08). Photosynthetic capacity ($V_{cmax}$ and $J_{max}$) estimates derived from measured leaf N content similarly exhibited a negative non-significant correlation with MCWD ($R^2$=0.51, p=0.07 and $R^2$=0.53, p=0.06 respectively). A positive non-significant correlation existed between model-calibrated leaf lifespan, measured LMA (log-log

$R^2$=0.39, p=0.14), and LAI ($R^2$=0.28, p=0.22). Model-calibrated leaf lifespan exhibited a negative non-significant correlation with photosynthetic capacity estimates ($V_{cmax}$ $R^2$=0.46, p=0.09; $J_{max}$ $R^2$=0.47, p=0.09). A significant positive correlation existed between mean annual LAI and LMA ($R^2$=0.85, p=0.003).

### 3.5 Model Experiments

### 3.5.1 Experiment 1. Drivers of Spatial Variation in GPP

Structural and trait responses to water availability explained more variation in GPP across the MCWD gradient than did climate. LAI accounted for the largest proportion of variance in mean annual GPP across plots (32.8%), whilst 21.3% was explained by differences in photosynthetic capacity (Table 6). Photosynthetic capacity increased with decreasing MCWD (Table 3); this relationship partially offset

the decrease in GPP linked to declining LAI. The direct effects of climate on GPP (which included physiological responses to water availability including stomatal conductance) accounted for 16.2% of plot variation in mean annual GPP. Rooting depth did not vary directionally with MCWD and consequently only had a small effect on GPP (4.1%). Soil properties and root biomass accounted for a very small fraction of variance (<2%).

### 3.5.2 Experiment 2. Variation in Forest Sensitivity to Drivers of GPP

The relative sensitivity of GPP to drivers varied across the MCWD gradient (Figure 6). GPP was most sensitive to changes in LAI (per unit $m^{-2}$ leaf area) for plots located at Caxiuanã, which experience the least negative MCWD and have large rooting depth (Caxiuanã LAI sensitivity Δ 537 gC $m^{-2}$ $yr^{-1}$ vs





overall mean LAI sensitivity $\Delta$ 380 gC m$^{-2}$ yr$^{-1}$). The sensitivity of GPP to LAI exhibited a positive,

non-significant correlation with MCWD (R$^2$=0.88, p=0.06; Tanguro LAI sensitivity $\Delta$ 286gC m$^{-2}$ yr$^{-1}$,

Kenia $\Delta$ 345 gC m$^{-2}$ yr$^{-1}$, Tambopata $\Delta$ 353 gC m$^{-2}$ yr$^{-1}$). Reflecting LAI trends, the sensitivity of GPP

to differences in photosynthetic capacity (per unit µmol C g s$^{-1}$) was similarly highest at Caxiuanã

(Caxiuanã photosynthetic capacity sensitivity $\Delta$ 27 gC m$^{-2}$ yr$^{-1}$, mean photosynthetic capacity sensitivity

$\Delta$ 20 gC m$^{-2}$ yr$^{-1}$; Table 3), and decreased linearly (though not significantly) across the MCWD gradient

(R$^2$=0.83, p=0.09; Tanguro photosynthetic capacity sensitivity $\Delta$ 16 gC m$^{-2}$ yr$^{-1}$, Kenia $\Delta$ 18 gC m$^{-2}$ yr$^{-1}$

$^1$, Tambopata $\Delta$ 18 gC m$^{-2}$ yr$^{-1}$). Tambopata plots, which have high LAI but shallow rooting depth, were

most sensitive to differences in climate (per unit MCWD mm) (3.44 gC m$^{-2}$ yr$^{-1}$), whilst Kenia plots,

which have similarly shallow rooting depth but low LAI, were the least sensitive (Kenia climate

sensitivity $\Delta$ 1.64 gC m$^{-2}$ yr$^{-1}$, Tanguro $\Delta$ 2.77 gC m$^{-2}$ yr$^{-1}$, Caxiuanã $\Delta$ 1.78 gC m$^{-2}$ yr$^{-1}$). The sensitivity

of GPP to differences in rooting depth (per m rooting depth) was highest at Tanguro and Tambopata

(Tanguro rooting depth sensitivity $\Delta$ 114 gC m$^{-2}$ yr$^{-1}$, Tambopata $\Delta$ 79 gC m$^{-2}$ yr$^{-1}$), and lowest at

Caxiuanã and Kenia (Caxiuanã rooting depth sensitivity $\Delta$ 28 gC m$^{-2}$ yr$^{-1}$, Kenia $\Delta$ 20 gC m$^{-2}$ yr$^{-1}$).

### 3.5.3 Experiment 3. Drivers of Sub-Annual Variation in GPP

In contrast to drivers of spatial variation in GPP, on a sub-annual timescale LAI had less explanatory

power than climate (Tables 6 and 7). The relative importance of solar radiation in driving monthly GPP

increased significantly with MCWD (R$^2$ = 0.90, p=<0.001), whilst the relative importance of LAI

declined (R$^2$=0.72, p=0.015). The relative importance of VPD did not vary directionally across the

MCWD gradient (R$^2$=0.10, p=0.49). Both precipitation and air temperature had little effect on monthly

GPP, though it is noted that a significant interaction existed between both precipitation and VPD

(p<0.001) and air temperature and shortwave radiation (p<0.001). Furthermore, temperature varied least

across plots in comparison to other climate forcings (standard deviation as a percentage of the mean;

temperature 9.8%, VPD 73%, precipitation 192%, shortwave radiation 34%). As such, seasonal changes

in the relative importance of temperature and precipitation were not investigated further. The relative

importance of LAI, VPD and solar radiation shifted seasonally, reflecting changes in the availability of

light and water. Solar radiation was typically the most important driver of monthly GPP during the wet



season, whilst VPD was more important during the dry season (Figure 7). The relative importance of LAI forcings peaked before dry season onset for forests under lower drought stress (Caxiuanã and Tambopata), and during the dry season for forests under higher drought stress (Kenia and Tanguro). Notably, LAI was also more important during the dry season at KEN02, which occupies shallow soil

(<1m) in comparison to KEN01.

## 4. Discussion

Our aim was to better understand the mechanisms coupling GPP and drought stress across Amazon forests. We found that, leaf traits (both modelled and observed) and LAI co-varied along the MCWD gradient. Across observed ranges in key variables, LAI was the principal driver of spatial variation in

GPP, followed by photosynthetic capacity (Q1). Forest sensitivity to differences in LAI and photosynthetic capacity decreased with increasing drought stress (Q2). Forests with higher evaporative potential (high LAI) relative to water supply were most sensitive to differences in climate and rooting depth. Solar radiation was a key driver of sub-annual variation in GPP, the relative effect of which increased with decreasing drought stress, coincident with declines in the relative importance of LAI,

consistent with the evaluation from the sensitivity analysis (Q3).

### 4.1 LAI and Leaf Traits along the MCWD gradient

Leaf trait parameters retrieved from SPA litterfall calibrations suggest a wide range of potential leaf lifespans across the MCWD gradient (~1-3 years), and are in accordance with estimates for Amazon tree species, reported by Reich et al. (1991) of between two months and four years (Table 4). Leaf trait

estimates co-varied across the MCWD gradient, in line with leaf economic theory (Wright et al., 2004). However, the interactions were often not significant. We suggest that in instances where $R^2$ values indicate a large proportion of variation is explained, high p-values may have occurred as a result of a small sample size (i.e. 7 plots). As drought stress increased, a shift towards deciduous strategies resulted in reduced leaf lifespan, but higher photosynthetic capacity. The co-variation of leaf traits along the

MCWD gradient shapes both the rate of carbon assimilation (via photosynthetic capacity), and the carbon economics of canopy dynamics (via LMA, leaf lifespan and metabolic rate). Coincident with





changes in leaf traits, mean annual LAI increased with decreasing drought stress. Whilst research efforts have focused on mapping LAI (Iio et al., 2014) and leaf trait (Kattge et al., 2011, Asner et al., 2015) distributions across climatic gradients, their covariance has not yet been explored. Given the role of leaf traits in shaping canopy carbon economics, the mechanisms underpinning LAI and leaf trait distributions across the resource availability gradient could prove important in understanding the effect of changes in precipitation regime on future Amazon carbon dynamics.

### 4.2 Drivers of Spatial Variation in GPP

Indirect effects of climate via ecosystem structure and long-term trait responses to water availability accounted for 54% of variation in GPP (Q1; Figure 1). Direct effects of climate (which included physiological responses to water availability) accounted for only 16% of observed variance (Table 6). Our results are consistent with previous reports on the importance of ecosystem structure and traits in determining spatial variation in GPP (Rodig et al., 2018, van de Weg et al., 2013, Reichstein et al., 2014), but go further to quantify the direct contribution of discrete drivers to observed variation in carbon assimilation. LAI explained the greatest proportion of variation in GPP, followed by photosynthetic capacity, whilst root and soil properties had little explanatory power.

Evidence of changes in LAI in response to precipitation regime has been presented across multiple ecosystems and over time (Grier and Running, 1977, Schleppi et al., 2011, Iio et al., 2014, Dobbertin et al., 2010, Wright et al., 2013). Amazonian forest throughfall exclusion experiments identified a decline in LAI with the onset of reduced soil water (Fisher et al., 2007, Meir et al., 2008, Brando et al., 2008). At Caxiuanã, over a 4-year period, observed leaf area was 20-30% lower than the control stand (Meir et al., 2009), with long-term reductions estimated at between 10-15% (Rowland et al., 2015a). Investigations show that declines in LAI are not caused by increased leaf turnover due to drought stress, but instead are the result of lower leaf production (Nepstad et al., 2002, Schuldt et al., 2011), suggesting an active response of plant allocation strategy to water availability. Concurrently, after 15 years under throughfall exclusion, Rowland et al. (2018) found that leaf litterfall still remained consistently lower. Reported trends in canopy dynamics are therefore in accordance with our findings, and indicate that LAI is a key response mechanism to precipitation regime. Whilst other studies such as da Costa et al.



(2018) have similarly pointed towards structural responses as the principal determinant of variation in

GPP, they identify changes in sapwood area as the main driver, rather than LAI. We suggest that whilst sapwood area may be more important in shaping the response to temporal changes in precipitation, for forests at steady state, emergent canopy properties (LAI) drive GPP trends.

Photosynthetic capacity also proved an important driver of spatial variation in GPP across the MCWD gradient. Our results are consistent with a number of Amazon-based studies linking leaf traits to

productivity (Aragao et al., 2009, Cleveland et al., 2011, Castanho et al., 2013). Interestingly, the observed shifts in photosynthetic capacity along the gradient had a compensatory effect on the GPP-MCWD interaction. Reductions in GPP under high drought stress were alleviated by higher photosynthetic capacitance. Similarly, shifts in photosynthetic capacity in response to temperature have been reported to reduce spatial variation in GPP across a tropical elevation gradient (Bahar et al., 2017,

van de Weg et al., 2013). Consistent with Fyllas et al. (2017), our results also show that the effect of climatic forcings on carbon fluxes can be successfully captured through spatial variation in canopy dynamics and leaf traits. However, as we have focused on the role of leaf traits in the absence of carbon cycle feedbacks, we do not take into account the effect of concurrent shifts in LMA and leaf lifespan, which together influence canopy carbon economics (Wright et al., 2004, Osnas et al., 2013, McMurtrie

and Dewar, 2011). Furthermore, as nutrient dynamics are not directly accounted for in SPA, we are unable to quantify the impact of soil nutrients on the GPP-MCWD interaction, beyond its manifestation in leaf traits. Nevertheless, the interaction between photosynthetic capacity and LAI proved important in driving variation in GPP across the MCWD gradient.

Root depth, root biomass and soil properties had little direct effect on spatial variation in GPP. Whilst

we recognise that the difficulty in measuring root depth and biomass (Metcalfe et al., 2007) adds uncertainty to our results, the findings do not reflect the importance of belowground functioning highlighted by other studies (Fisher et al., 2007, Metcalfe et al., 2008, Baker et al., 2008, Phillips et al., 2009, Ichii et al., 2007). Notably, a number of GEM plots had hard pan layers (Quesada et al., 2012) so may be acclimated to operate in shallow rooting zones, and are therefore not necessarily representative

of other Amazon forests under the same precipitation regime. However, it is likely that given these



drivers are largely associated with the acquisition of water, rather than carbon, if feedbacks were enabled within analyses, root and soil properties would prove to have a stronger effect.

### 4.3 Variation in Forest Sensitivity to Drivers of GPP

The sensitivity of GPP to differences in LAI, climate, photosynthetic capacity and rooting depth varied
across the MCWD gradient with evaporative potential and water uptake capacity (Q2; Figure 6). As the model experiment was conducted in the absence of carbon cycle feedbacks, sensitivities reflect shorter rather than long-term effects of changes in forcings. The sensitivity of GPP to differences in LAI and photosynthetic capacity was greatest for forests occupying the lowest drought stress zone and declined with increasing drought stress. Our results are in agreement with findings from Wright et al. (2013),
who reported that GPP was most sensitive to decreases in leaf area when water availability was highest. Forests with a high LAI (and therefore high evaporative potential) but shallow rooting depth were most sensitive to differences in climate. Our results suggest that where rooting depth is relatively shallow, and unable to ameliorate the effects of drought stress as seen elsewhere (Nepstad et al., 2007, Malhi et al., 2009a), forests with a high LAI could be more vulnerable to reduced precipitation. Investigations
into the vulnerability of Amazon forests to drought have put a deservedly large emphasis on the role of physiological responses (Choat et al., 2012, Phillips et al., 2009, Bennett et al., 2015, Corlett, 2016). However, our results indicate that the role of ecosystem structure could also prove important, and that forests with a high evaporative potential (high LAI) but low water uptake capacity (shallow rooting depth) should be a focus for future studies.

### 4.4 Drivers of Sub-Annual Variation in GPP

Seasonal variation in GPP was driven by changes in solar radiation, VPD and LAI, the relative importance of which, was dependent on MCWD (Q3; Figure 7). Shortwave radiation was the dominant driver of sub-annual variation in GPP across plots, and its relative effect increased with decreasing drought stress (Table 7). The relative importance of LAI in driving sub-annual GPP increased with
drought stress. A number of studies report that for Amazon forests subject to significantly low annual rainfall, GPP declines with increased VPD, and in accordance with our findings, in higher rainfall zones





GPP increases in line with solar radiation (Von Randow et al., 2013, Goulden et al., 2004, Hutyra et al., 2007, Saleska et al., 2003, Rowland et al., 2014, Carswell et al., 2002). Our results suggest that LAI is not the principal driver of sub-annual variance in GPP, in contrast to its role in driving spatial variation

across the MCWD gradient. However, whilst other studies agree that leaf area alone does not drive variation in sub-annual GPP (Wu et al., 2017, Wu et al., 2016, Brando et al., 2010, Restrepo-Coupe et al., 2013, Bi et al., 2015), we fail to account for shifts in photosynthetic capacity with leaf age. The coordination of leaf flushing and new leaf emergence with climatic drivers such as solar radiation is thought to exceed the effects of LAI in non-water limited forests (Myneni et al., 2007). We further

recognise the uncertainty introduced through using leaf N content to derive photosynthetic capacity estimates (for five of the seven plots), given the distribution of leaf N between photosynthetic and non-photosynthetic proteins is not fixed (Onoda et al., 2017). However, notwithstanding the effects of temporal variation in photosynthetic capacity, we demonstrate the relative importance of canopy dynamics and climatic forcing driving variation in GPP, shift with light and water availability. Our

results indicate that with respect to soil moisture, GPP is demand limited across spatial scales, but is supply limited across sub-annual timescales.

## 4.5 Limitations and Opportunities

Given the importance of LAI in driving variation in GPP, data on canopy dynamics is critical to constrain carbon flux estimates across the Amazon basin. Our approach utilised field estimates of LAI

from hemispherical photographs to constrain model simulations. The accuracy and spatial validity of indirect estimates of LAI has been questioned at higher leaf areas (Bréda, 2003, Jonckheere et al., 2004, Weiss et al., 2004). In this study, we expect that if field measurements of LAI were underestimated at higher leaf areas, the proportion of spatial variation in GPP explained by LAI would increase, as a result of increased variation in both field-measured and model simulated GPP. Yet, our highest estimates of

LAI (Caxiuanã $5.11 \pm 1.41$ $m^2m^{-2}$) align with destructive sampling measurements from a terra-firme Amazon forest (McWilliam et al. (1993) $5.7 \pm 0.5$ $m^2m^{-2}$). Furthermore, a comparison of LAI estimation approaches (Asner et al., 2003) suggested that indirect methods were appropriate for broadleaved forests, and presented no statistical difference between destructive harvesting and indirect methods.



However, the use of ground measurements is limited to smaller spatial scales, and LAI estimates across
the basin are needed to constrain carbon flux estimates. Whilst the interpretation of forest responses to
drought stress through remote sensing approaches have caused controversy (Asner and Alencar, 2010,
Saleska et al., 2007, Samanta et al., 2010), an increase in canopy mapping through satellite missions
could be instrumental to efforts aiming to better understand LAI dynamics. Current and upcoming
satellite missions including FLEX (FLuorescence EXplorer), GEDI (Global Ecosystem Dynamics
Investigation) and Sentinel will offer opportunity for new insights into changes in leaves *in-situ*, vertical
canopy structure, and temporal variability via repeat measurements (Morton, 2016, Drusch et al., 2017,
Pettorelli et al., 2018). Efforts to map trait distributions will also prove important (Kattge et al., 2011,
Asner et al., 2015) given their role in driving variation in GPP.

## 5. Conclusion

We show that indirect effects of climate (via ecosystem structure and trait responses) exceed direct
effects (via physiological responses) in driving spatial variation in GPP across an Amazon MCWD
gradient (Q1). Conversely, across sub-annual timescales, the reverse was true (Q3). The relative
sensitivity of GPP to changes in direct and indirect forcings shifted across the MCWD gradient and was
dependent on water availability, demand and acquisition potential (Q2); identifying the potential
vulnerability of forests with a high evaporative potential (i.e. high LAI), but low water uptake capacity
(i.e. shallow rooting depth), to changes in precipitation regime. Given the role of LAI in driving GPP
across the drought stress gradient, we highlight a requisite for improved mapping of canopy dynamics
(via remote sensing). We propose that ecosystem model development should focus on the integration
of structural and trait responses to drought stress (alongside physiological responses). The inclusion of
both direct and indirect effects of climate in ecosystem models, would reduce current uncertainty in
predicted annual and sub-annual GPP for tropical forests.
**Supplementary Material**

Supplementary material is included in a separate document.

**Authorship Contributions** Sophie Flack-Prain, Mathew Williams and Patrick Meir conceived the research questions. Data used in model calibration and validation was collected by Yadvinder Malhi and associates. Model experiments were designed and conducted by Sophie Flack-Prain with

contributions from Mathew Williams and Thomas L. Smallman. Sophie Flack-Prain and Mathew Williams prepared the manuscript with active contributions from all co-authors.

**Conflict of Interest Statement:** There are no conflicts of interest to disclose.

**Acknowledgements:** The authors would like to thank the Global Ecosystems Monitoring network team for the field data used in this study, collected through funding from NERC and the Gordon and Betty

Moore Foundation, and an ERC Advanced Investigator Award to YM (GEM-TRAIT). The authors would also like to thank the PhD project funding body, the UK Natural Environment Research Council E3 DTP, the National Centre for Earth Observation, the UKSA project Forests 2020 and the Newton CSSP Brazil. The TRY trait database is thanked for the data used in model parameterisation.





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



Tables

Table 1. Environmental characteristics of GEM network Amazon permanent sample plots across the
MCWD gradient. Meteorological data is from local weather stations, gap filled with ERA interim data
for the years 2009-2010 (Dee et al., 2011).

| Plot name | Caxiuanã Control | Caxiuanã Tower | Tambopata V | Tambopata VI | Kenia Wet | Kenia Dry | Tanguro Control |
|---|---|---|---|---|---|---|---|
| RAINFOR site code | CAX04 | CAX06 | TAM05 | TAM06 | KEN01 | KEN02 | --- |
| Latitude | -1.716 | -1.737 | -12.831 | -12.839 | -16.016 | -16.016 | -13.077 |
| Longitude | -51.457 | -51.462 | -69.271 | -69.296 | -62.73 | -62.73 | -52.386 |
| Elevation (m.a.s.l) | 47 | | 223 | | 384 | | 385 |
| Mean Maximum Climatological Water Deficit (mm) | -85.5 | | -256 | | -342 | | -498 |
| Mean annual air temperature ($^0$C) | 26.1 | | 24.6 | | 23.4 | | 25.4 |
| Soil Type | Vetic Acrisol | Ferralsol | Cambisol | Alisol | Cambisol | Cambisol | Ferralsol |
| Soil N (%) | 0.06 | 0.13 | 0.16 | 0.17 | 0.22 | 0.17 | 0.16 |
| Soil $P_{total}$ (mg kg$^{-1}$) | 37.4 | 178.5 | 256.3 | 528.8 | 447.1 | 244.7 | 147 |






Table 2. Summary of the relationship between model variables and field data. Values are either a SPA
model parameter (input) or output. Model parameters may be initial conditions subsequently allowed

to fluctuate, a fixed value, or a time-series, whereby the parameter value at each time point is prescribed
to the model. Model outputs are generated on either an hourly or daily time-step and are presented in
the text as the mean annual sum (2009-2010), unless otherwise stated. Model outputs are calibrated or
evaluated using field data. Values are specific to each of the seven GEM Amazonian permanent sample
plots.

| Value | Model Parameter or Output | Source of Value or Calibration/Validation Data |
|---|---|---|
| LMA | parameter (single fixed) | GEM plot-measured value or literature-based estimate from plot species list |
| $V_{cmax}$ | parameter (single fixed) | (estimate from) GEM plot-measured value or TRY database estimate from plot species list |
| $J_{max}$ | parameter (single fixed) | (estimate from) GEM plot-measured value or TRY database estimate from plot species list |
| Leaf N content | parameter (single fixed) | GEM plot-measured value or TRY database estimate from plot species list |
| LAI | parameter (timeseries fixed) | GEM monthly plot-measured value |
| Leaf NPP | output | model calibration to GEM plot-measured leaf litterfall and LAI |
| Wood NPP | | |



| | | |
|---|---|---|
| *fraction of total NPP* | parameter (single fixed) | GEM plot-measured value |
| *total wood NPP* | output | simulated value validated against GEM field-measured total wood NPP |
| **Root NPP** | | |
| *fraction of total NPP* | parameter (single fixed) | GEM plot-measured value |
| *total root NPP* | output | simulated value validated against GEM field-measured total root NPP |
| **Leaf turnover** | parameter (single fixed; function of three individual fixed parameters relating to the leaf litterfall cycle) | model calibration to GEM plot-measured leaf litterfall |
| **Root turnover** | parameter (single fixed) | estimated using root NPP assuming steady state conditions |
| **Wood turnover** | parameter (single fixed) | estimated using wood NPP assuming steady state conditions |
| **Foliar C stock** | parameter (timeseries fixed) | product of LAI and LMA |




| | | | |
|---|---|---|---|
| **Wood C stock** | parameter | initial condition; thereafter output | initial condition uses GEM plot-measured DBH values converted to C stock using allometric equation |
| | | | output calculated in SPA as simulated wood C stock plus NPP minus turnover |
| **Root C stock** | parameter | initial condition; thereafter output | initial condition used GEM plot-measured root stock values or literature-based estimate |
| | | | output calculated in SPA as simulated root C stock plus NPP minus turnover |
| **Leaf respiration** | output | | sum of leaf maintenance and growth respiration; maintenance respiration generated using measured leaf N content, foliar C stock and the Reich *et al*., (2008) leaf respiration model, validated against GEM estimates; growth respiration calculated in SPA as leaf NPP × 0.28 |
| **Wood respiration** | output | | sum of wood maintenance and growth respiration; maintenance respiration calculated as a function of wood C stock, the coefficient being derived from GEM estimates; growth respiration calculated in SPA as wood NPP × 0.28 |
| **Root respiration** | output | | sum of root maintenance and growth respiration; maintenance respiration calculated as a function |



| | | of root C stock, the coefficient being derived from GEM estimates; growth respiration calculated in SPA as root NPP × 0.28 |
|---|---|---|
| **Respiration** | output | sum of simulated leaf, wood and root respiration, evaluated against GEM data |
| **GPP** | output | generated through SPA process-based modelling of GPP using detailed parameters, evaluated against GEM data |
| **NPP** | output | calculated in SPA as GPP minus autotrophic respiration, evaluated against GEM data |





Table 3. Field estimated mean annual leaf area index (LAI), leaf traits, maximum rooting depth and fine root biomass for Amazon permanent sample plots along a MCWD gradient. LAI estimates were derived from monthly hemispherical photographs. LAI, leaf trait and rooting depth estimates were used to constrain SPA model runs. Estimate standard errors are presented in brackets. Fine root C stock estimates were absent for Tanguro plots.

|  | LAI ($m^2\ m^{-2}$) | LMA ($g\ m^{-2}$) | leaf N content ($g\ m^{-2}$) | Maximum Rooting Depth (m) | Fine Root C Stock ($g\ C\ m^{-2}$) |
|---|---|---|---|---|---|
| **CAX04** | 4.99 (± 1.07) | 93 (± 17) | 1.82 (± 0.43) | 8 | 345 |
| **CAX06** | 5.23 (± 0.92) | 87 (± 54) | 2.12 (± 0.7) | 10 | 433 |
| **TAM05** | 4.85 (± 0.81) | 101 (± 24) | 2.38 (± 0.56) | 1 | 770 |
| **TAM06** | 4.64 (± 0.77) | 96 (± 21) | 2.51 (± 0.64) | 1 | 500 |
| **KEN01** | 2.77 (± 0.17) | 53 (± 13) | 2.12 (± 0.25) | 2 | 818 |
| **KEN02** | 2.22 (± 0.14) | 42 (± 13) | 2.31 (± 0.31) | 1 | 607 |
| **Tanguro** | 4.13 (± 1.01) | 64 (± 13) | 2.01 (± 0.52) | <10 | - |







Table 4. SPA calibrated leaf litterfall parameters for plots across an Amazon MCWD gradient. Peak

leaf fall is the day of year leaf litterfall reaches its maximum, leaf lifespan reflects maximum lifespan

of leaves and leaf fall period is the number of days over which systematic increases in leaf fall occur.

Leaf litterfall parameters were calibrated against GEM field estimates to capture leaf litterfall and

timing.

|  | Peak Leaf Fall | Leaf Lifespan | Leaf Fall Period |
| --- | --- | --- | --- |
|  | (day of year) | (years) | (days) |
| **CAX04** | 210 | 3.00 | 150 |
| **CAX06** | 190 | 1.45 | 100 |
| **TAM05** | 220 | 1.30 | 130 |
| **TAM06** | 230 | 1.42 | 100 |
| **KEN01** | 200 | 1.05 | 100 |
| **KEN02** | 180 | 1.01 | 100 |
| **Tanguro** | 180 | 1.04 | 120 |







Table 5. A comparison of GEM field measurements and SPA process-based modelling estimates of component autotrophic respiration and NPP. We present the $R^2$, p-value, and root mean square error (RMSE) of the interaction between SPA and GEM annual estimates, together with the range in GEM biometrically derived estimates across seven sample plots at four locations in the Amazon basin.

| Component | $R^2$ | p-value | RMSE | Range in Field Estimates (gC m$^{-2}$ yr$^{-1}$) |
|---|---|---|---|---|
| **Respiration** | | | | |
| Foliage | 0.47 | 0.09 | 92.0 | 454-830 |
| Wood | 0.75 | 0.01 | 100.5 | 411-1054 |
| Fine Root | 0.91 | <0.001 | 74.1 | 232-1041 |
| **NPP** | | | | |
| Foliage | 0.99 | <0.001 | 9.0 | 150-491 |
| Wood | 0.21 | 0.30 | 25.3 | 189-292 |
| Fine Root | 0.59 | 0.04 | 49.5 | 189-418 |





Table 6. The proportion of variation in GPP across seven GEM Amazonian permanent sample plots explained by photosynthetic drivers in SPA. Model drivers were alternated individually at each plot to that of all other plots and the resultant change in GPP retrieved. Proportion of variance explained was calculated as conditional sum of squares divided by the total sum of squares (n=476; where the conditions were LAI, photosynthetic capacity, rooting depth, root biomass, climate and soil).


| Driver | Percentage of Variation Explained (%) |
|---|---|
| LAI | 32.8 |
| Photosynthetic capacity | 21.3 |
| Climate | 16.2 |
| Root depth | 4.1 |
| Soil | 1.2 |
| Root biomass | 0.7 |







Table 7. The relative importance of LAI, VPD, solar radiation, precipitation and air temperature ($T_{air}$) in driving monthly variation in GPP (%). Monthly GPP estimates are derived from calibrated SPA simulations for seven permanent sample plots across an Amazon MCWD gradient, constrained using monthly field LAI measurements. Relative importance values were derived from analyses using the random forest technique (n=168).

| Plot | LAI | VPD | Solar Radiation | Precipitation | $T_{air}$ |
|------|-----|-----|-----------------|---------------|-----------|
| CAX04 | 13 | 17 | 58 | 8 | 5 |
| CAX06 | 6 | 16 | 64 | 8 | 5 |
| TAM05 | 17 | 22 | 53 | 3 | 5 |
| TAM06 | 17 | 21 | 53 | 3 | 7 |
| KEN01 | 16 | 21 | 45 | 10 | 8 |
| KEN02 | 32 | 14 | 42 | 4 | 8 |
| Tanguro | 33 | 20 | 24 | 6 | 10 |








Figures

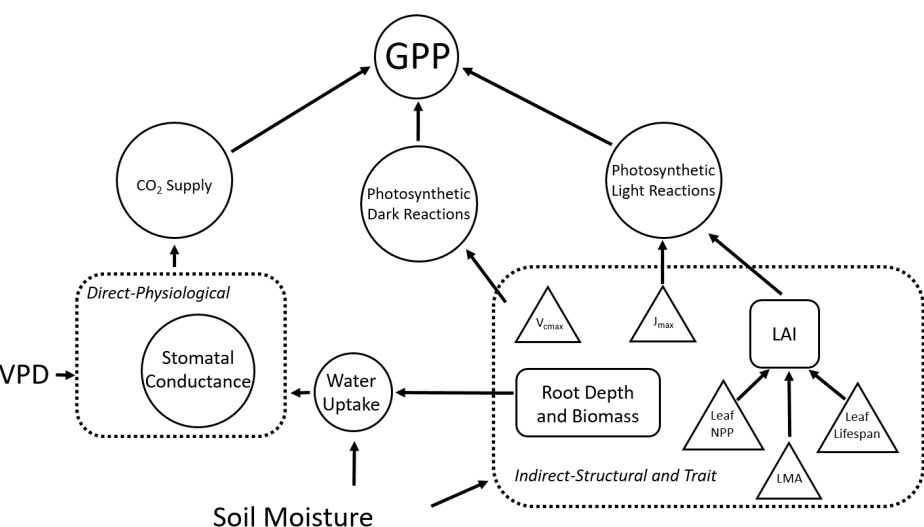

Figure 1. A schematic of the direct and indirect effects of drought stress (via soil moisture and VPD on

GPP. Drought stress affects GPP directly via stomatal conductance, and indirectly through its

determinant effect on plant traits and structural properties. Plant processes are represented by circles,

traits are represented by triangles and vegetation properties (i.e. ecosystem structure) are represented

by rectangles. Dashed boxes identify interactions driving the direct and indirect pathways through

which drought stress impacts GPP. We note that other climate forcings (e.g. shortwave radiation and

temperature) impact GPP but are not included here.




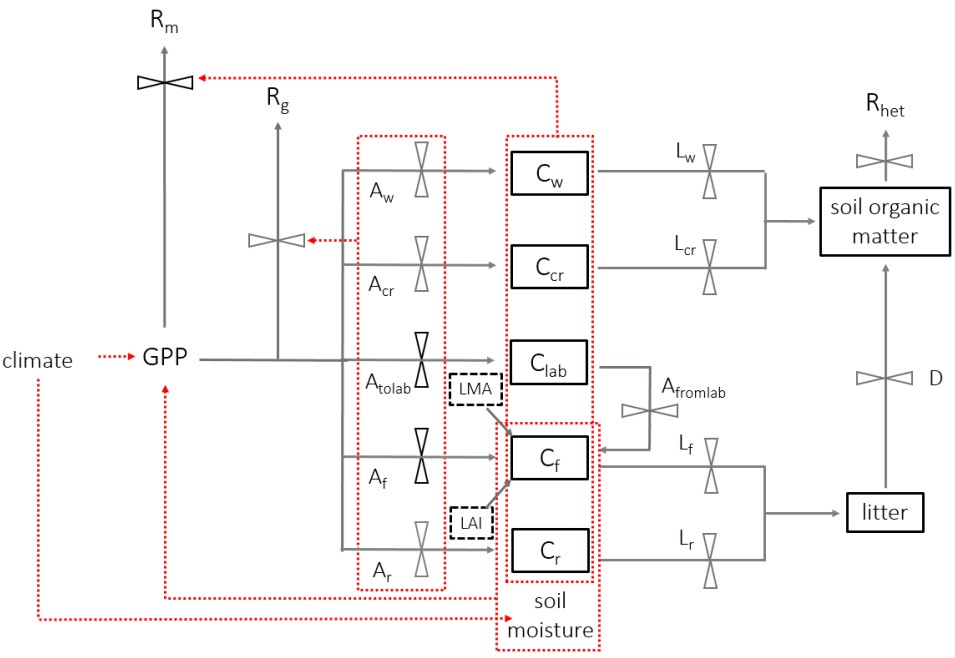

Figure 2. A schematic of DALEC$_{canopy}$, the carbon allocation sub-model integrated within the soil-plant-atmosphere model. Carbon moves between pools (solid boxes) via fluxes (solid arrows). Leaf carbon fluxes are constrained by field measurements (black dashed boxes). An effect of climate, carbon pools or fluxes on another carbon flux is shown by a red dashed arrow, whereby red dotted boxes indicate a collective impact of the contained carbon pools or fluxes. Black flux bars indicate that the carbon pathway is prioritised within the model above pathways from the same nodule. Climate is a model input, whilst soil moisture is simulated within SPA. Carbon pools (C), allocation (A) and litterfall (L) are separated by component: w = wood, cr = coarse roots, r = fine roots, f = foliage, lab = labile (or non-structural carbon), with to and from used for labile carbon.

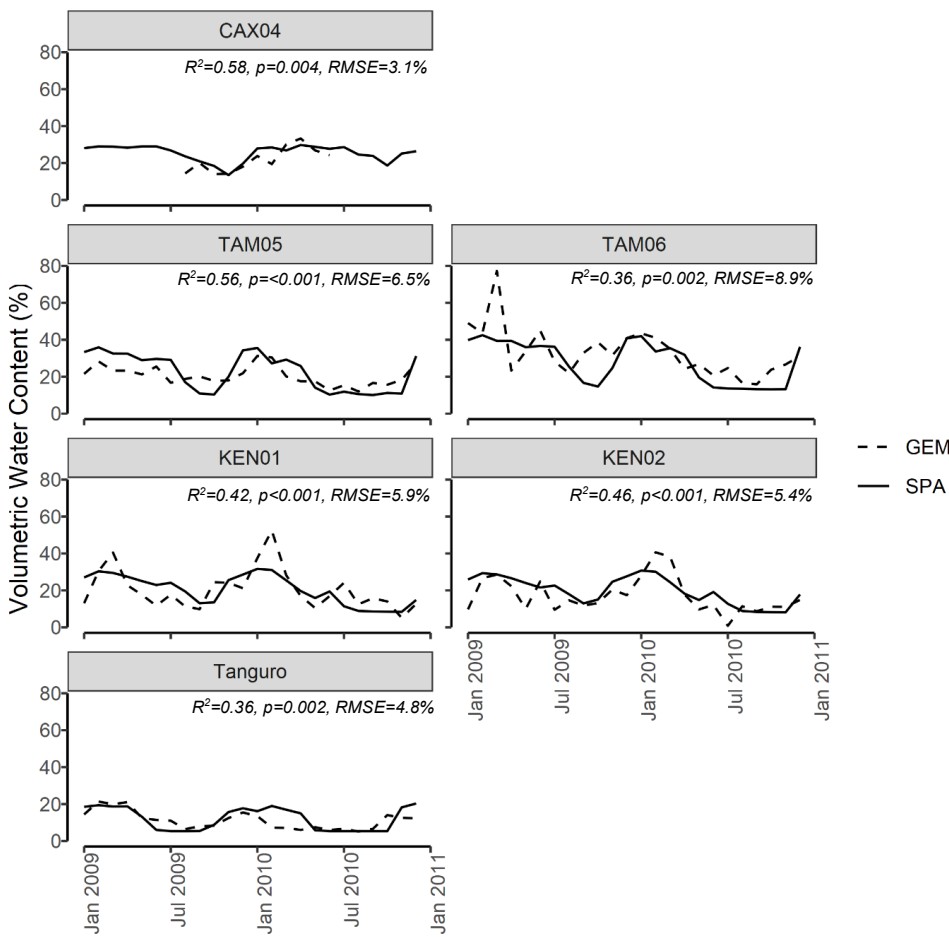

Figure 3. SPA estimated soil volumetric water content compared to GEM measured values for six of

the seven sample plots at four locations across the Amazon basin. Data presented is for the time period

2009-2010. Field data for CAX04 was limited to a shorter time period and for CAX06 was unavailable.

$R^2$, p-value and RMSE estimates presented are derived from linear regressions between monthly GEM

measurements and SPA simulations.



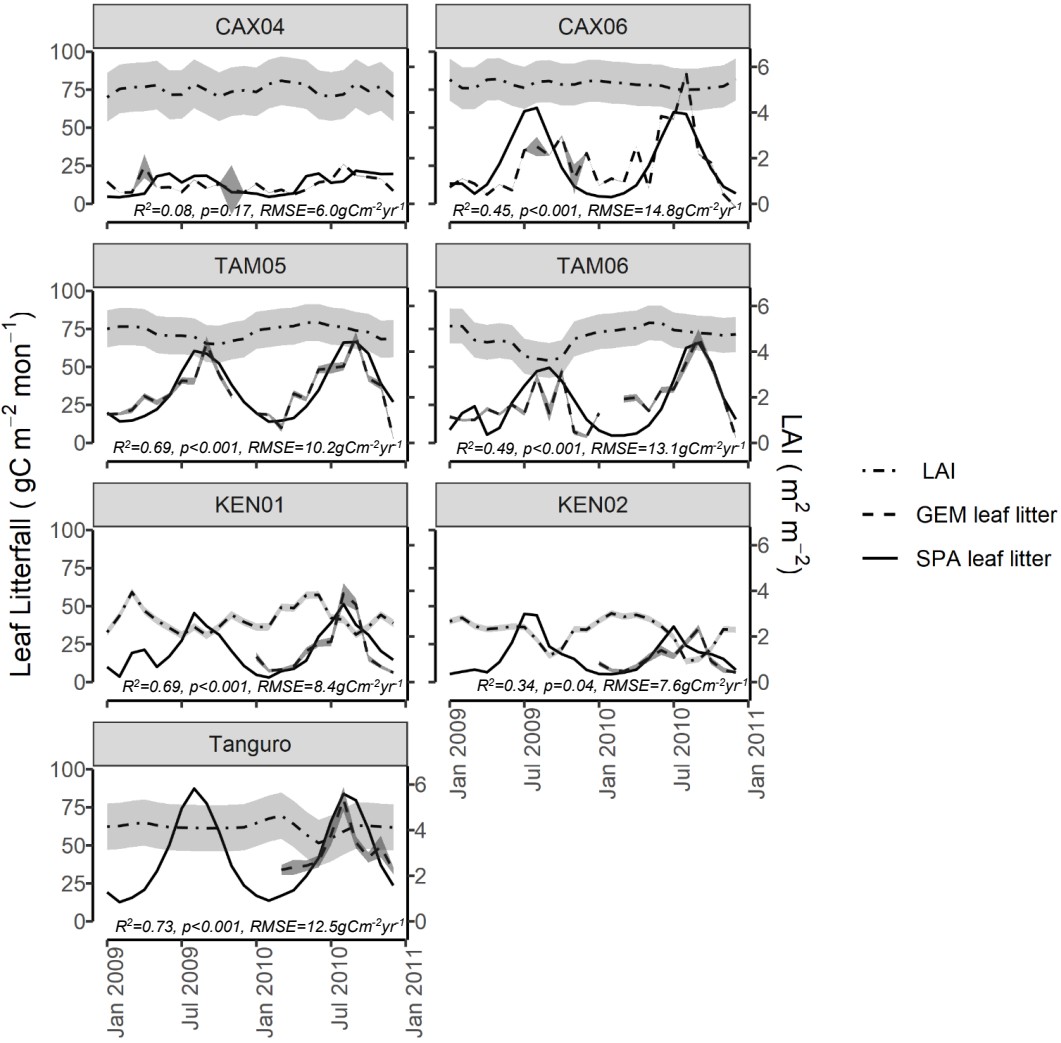

Figure 4. Field estimated monthly LAI, leaf litterfall (GEM), and standard error, compared with SPA simulated leaf litterfall for seven plots at four locations across the Amazon basin. SPA leaf litterfall was calibrated against GEM estimates to derive three fixed model drivers relating to the leaf cycle (peak leaf fall timing, leaf fall period and leaf lifespan). GEM leaf litterfall data was available for 2009-2010 for CAX04, CAX06, TAM05, TAM06 and for 2010 only for KEN01, KEN02 and Tanguro. $R^2$, p-value and RMSE estimates presented are derived from linear regressions between monthly GEM measurements and SPA simulations.

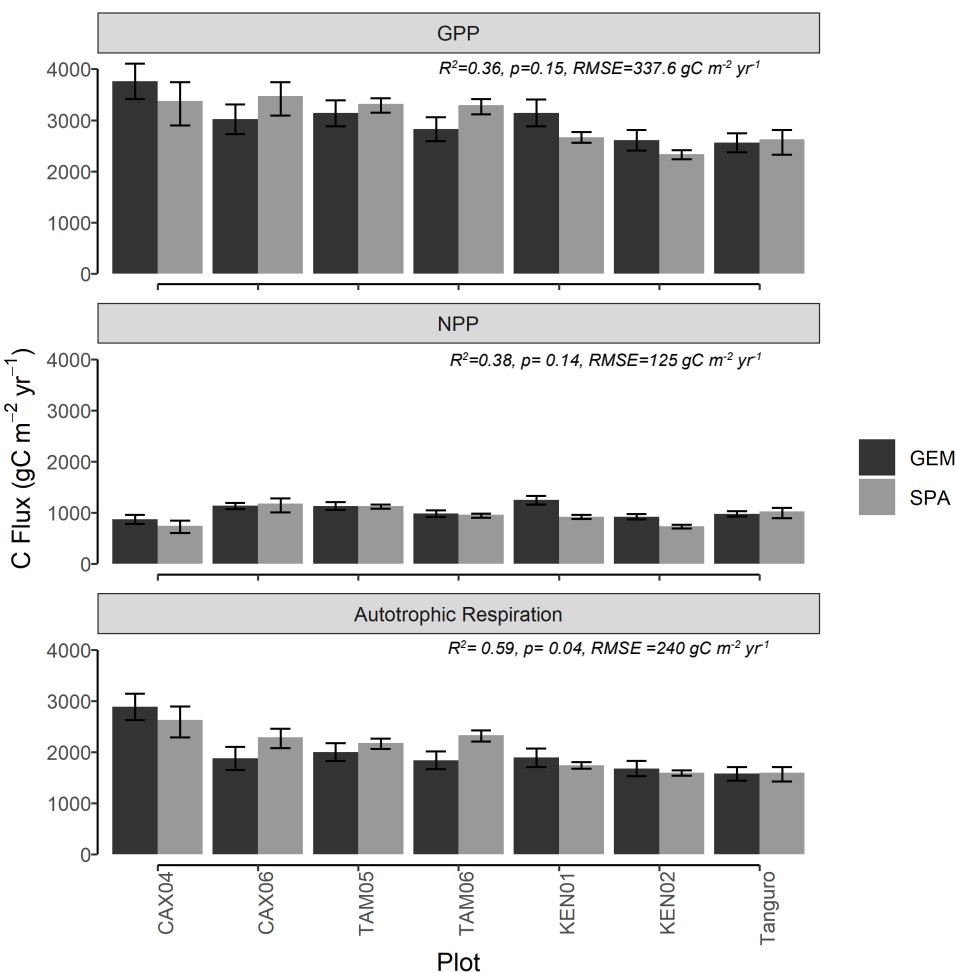

Figure 5. Carbon flux estimates (gC m⁻² yr⁻¹) of (a) GPP, (b) NPP and (c) autotrophic respiration, derived

from process-based modelling (SPA) and biometric methods (GEM) for seven permanent sample plots

at four locations across the Amazon basin. Estimates are mean annual values representative of the years

2009-2010. GEM error bars represent standard error from field carbon flux measurements. SPA error

bars represent simulated C fluxes under the upper and lower field LAI standard error. $R^2$, p values and

RMSE represent the interaction between SPA and GEM C flux estimates.



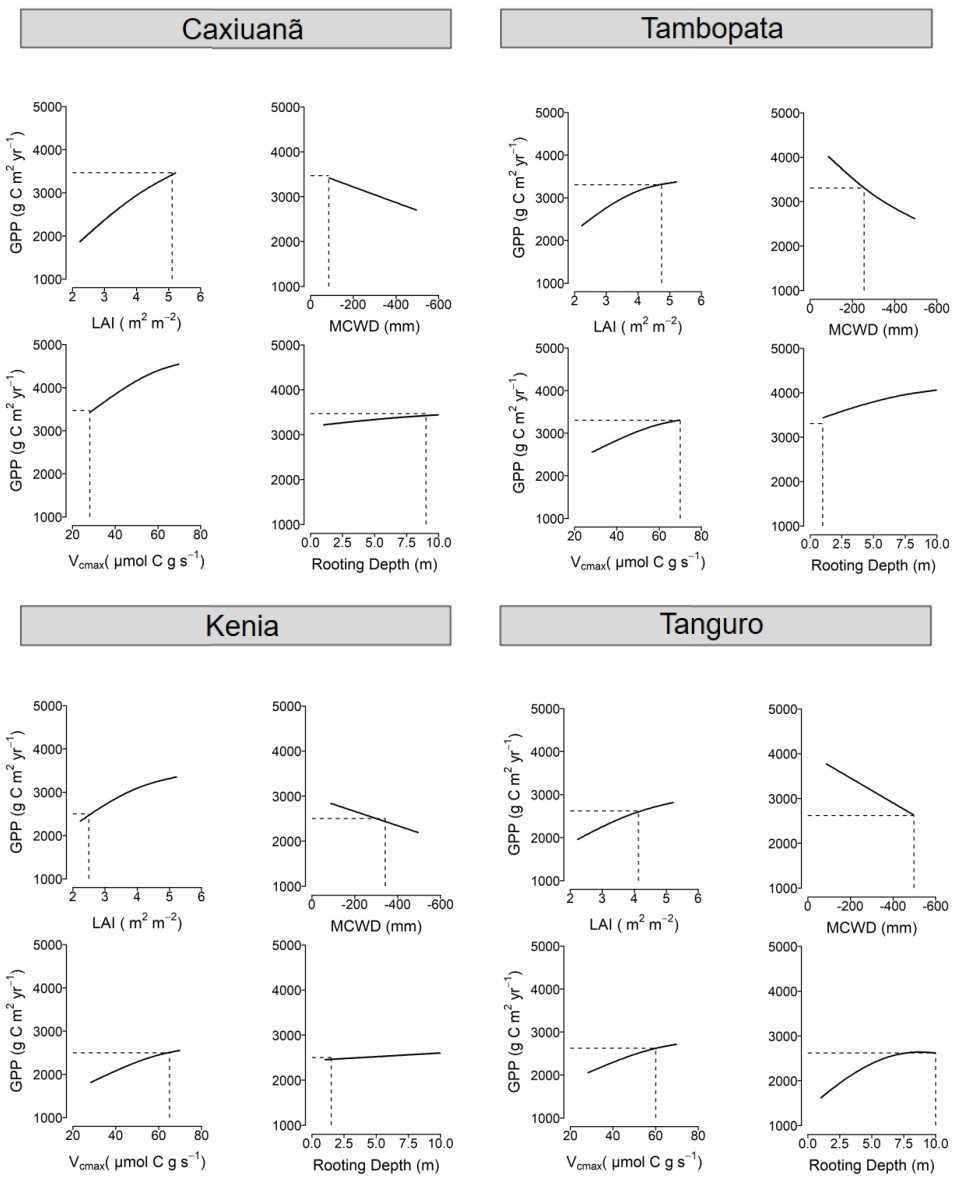

Figure 6. The sensitivity of GPP to model driver alternations in SPA at each location. Model drivers

LAI, climate (characterised by MCWD), photosynthetic capacity (characterised by $V_{cmax}$) and rooting

depth, derived from field observations, were alternated individually at each plot to that of all other plots

and the resultant GPP retrieved. Solid lines represent SPA simulated GPP under the named driver

alternations, whilst the dashed line represents the simulated value under observed conditions. SPA GPP



estimates presented are location averages. Climate and LAI were input to the model as timeseries, whilst

photosynthetic capacity and rooting depth were fixed values. Plots are ordered to reflect soil moisture-

stress which increases from Caxiuanã >Tambopata>Kenia>Tanguro.






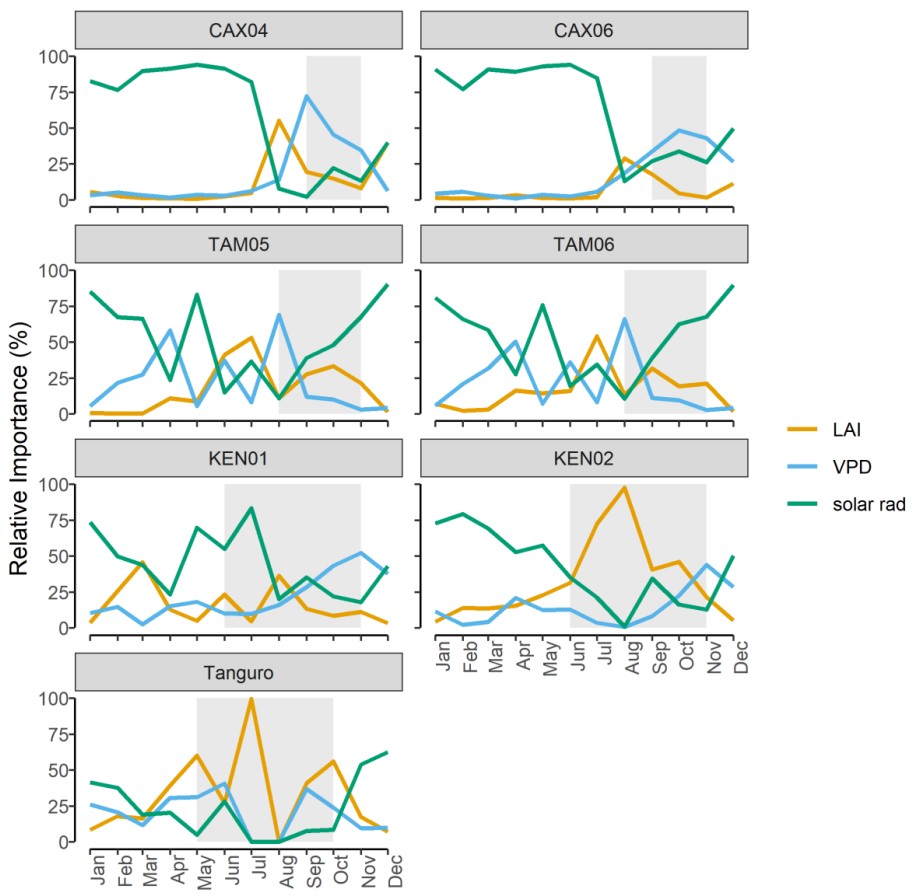

Figure 7. The relative importance (%) of LAI, vapour pressure deficit (VPD) and solar radiation (solar rad) in driving SPA estimated monthly photosynthesis at permanent sample plots across an Amazon MCWD gradient. Relative importance was calculated using random forest machine learning. Shaded regions represent the dry season, where monthly precipitation was below 100mm. Plots are ordered to reflect drought stress which increased from Caxiuanã> Tambopata> Kenia> Tanguro.