# Peer review of "The Importance of Physiological, Structural and Trait Responses to Drought Stress in Driving Spatial and Temporal Variation in GPP across Amazon Forests Sophie Flack-Prain1, Patrick"

_Biogeosciences, 2019_

## Referee Comment (RC1) · Anonymous Referee #1 · 18 Jun 2019

**Referee comments**

The authors investigate the importance of different drivers (LAI, leaf traits, climate) for GPP at both temporal and spatial scale across a drought stress gradient in the amazonian region using the Soil-Plant-Atmosphere (SPA) ecosystem model. The SPA model is applied at 7 sites, using sites specific parameters and is forced with LAI observed from hemispherical photographs. Simulation experiments and machine learning techniques are used to investigate their scientific questions. They find that indirect effects via plant traits and ecosystem structural changes, here expressed as LAI, are found to be the main driver of GPP across a spatial drought gradient, but the sensitivity of GPP to changes in these drivers varied with the gradient. On a sub-annual timescale climatic drivers were found to be more important for GPP. The authors discuss how these

direct physiological and indirect mechanism affect GPP but fail to explain the added value of forcing their model with observed LAI and to explain in detail how this forced LAI propagates down the modelling structure of SPA. The manuscript is well written and well structured, however, with many repetitions that should be deleted to make space for more details on your methods. As explained in detail in the comments below, I would like the authors to consider my questions and comments, before I recommend the publication of this manuscript.

**General comments**

- Throughout the manuscripts, your hypothesis/conclusions are repeated (abstract, introduction, discussion, and conclusion). This takes up a lot of space that could otherwise have been used elsewhere in the manuscript. Therefore, I urge you to delete several of these repeated paragraphs. Please see the specific comments below for my suggestions.

- The title *Leaf Area Index Changes Explain GPP variations across an Amazon Drought Stress Gradient* is not surprising as LAI generally scales well with GPP, and hence you would expect the changes to do so as well. Moreover, as also stated in the manuscript, the changes of LAI are affected by drought stress, and thus it is indirectly the drought stress that is causing the variation in GPP. Lastly, the title does not fully cover all three research questions made by the authors in the manuscript, although it points towards your most interesting finding. However,I would suggest that you reconsider the title.

- Several times you state that changes in LAI is an indirect structural effect from changes in soil moisture. From there, it follows that it is LAI which drives the GPP across the MCWD gradient. A strong emphasis is throughout the manuscript put on LAI and LAI as a driver of GPP, while LAI is strongly impacted by drought stress. However, the model is forced with LAI from hemispherical photographs,

but the authors do not explain how the forced LAI is linked to and impact the simulated soil moisture content. From Fig. 2 it follows that LAI impacts the foliage carbon pool, and this pool together with carbon pool of fine roots and soil moisture impacts GPP, but the link between the forced LAI and soil moisture is not well explained for your model setup. Please clarify this in the manuscript.

- Several times you briefly mention the C allocation (line 232 to 241, Fig. 2 and Supplement material). In the text you state that allocation to $NPP_{leaf}$ occurs first. Normally NPP is considered a flux, and normally you would allocate to a pool. Thus, do you mean that allocation to the foliar stock occurs first? If assimilation does not provide the C need for allocation to support the LAI, you take from the labile/non-structural carbon pool. However, in the supplement material in the last three equations, you state that if the labile pool has been depleted you allocate from the total NPP. Surely this must only be the case when you have enough NPP to sustain the foliar stock as required by the LAI. Please clarify this in the manuscript.

**Specific comments:**

Line 88 can be read as if you say LAI is a trait. Conventionally, LAI is not considered a trait (you could use max LAI), but rather relates to the ecosystem structure. Thus, for clarification could you please consider rephrasing the sentence to e.g.: *Changes to both ecosystem structure and traits, such as LAI, rooting depth and carboxylation capacity, are expected to be more longstanding (Meir et al., 2015a).*

Line 142-162 As the introduction is already very long, and much of your hypothesis is repeated later in the manuscript, I would highly recommend deleting these paragraphs.

Line 175-181 These lines are almost identical to your scientific research questions listed in the introduction. Please consider deleting one or the other.

Line 232-234 You state that the mapping of canopy dynamics is critical, and that

changes in canopy dynamics cause disparity between field observations and model predictions – how well is canopy dynamics simulated by SPA? How is the LAI forced over the canopy layers in SPA? Please elaborate on these aspects in the manuscript and explain how your study improve these shortcomings.

Line 232-237 NPP$_{Leaf}$ was calculated as the difference between the foliar C stock of the previous time step and that which would equate to field measured LAI. The field measured LAI has a monthly resolution. In principle you would have foliar C stock that could change at every model time step. But if the foliar C stock already equates to the field measured LAI, because the resolution of the forced LAI is monthly, NPP$_{Leaf}$ would just be zero.

Line 239 This sentence is not clear.

Line 243 How was the SPA model calibration constraint by an upper and lower sample error of LAI? Please add clarification to the text.

Line 301-302 These lines are repetitions, and not needed. Please consider deleting.

Line 308 There is something wrong with the structure of this sentence.

Line 320-325 The correlation between GEM estimated and SPA simulation GPP are non-significant and moderate. As GPP is imperative for your analysis, have you considered the impact it might have on you results? Have you investigated how the GEM estimated GPP relates to the LAI form the hemispherical photographs?

Line 326 Please explain why the GPP$_{SPA}$ variance is calculated under the LAI standard error.

Line 401-410 This paragraph sounds like a conclusion, and since you have a Conclusion section, where this is also stated, I would suggest you delete this paragraph.

Line 437-449 You mention how changes in LAI is a response to precipitations regimes, and even call it a key response mechanism. Then, one could infer that it is just as

much changes in precipitation that explains the changes in GPP as it is LAI. You state that changes in LAI is an indirect effect from climate — although the impact might be somewhat delayed, but did you investigate lagged correlations and variances between precipitation and GPP?

Line 452 You do not have a forest at steady state, if it is changing and experiencing trends. Please explain what you mean by a steady state forest – is it continuous cover, constant number of trees etc.? Or better yet refrain from using the term.

Line 465-468 With the SPA model you are not able to quantify the impact of soil nutrient on the GPP-MCWD interactions because you lack nutrient cycling in the SPA model. However, according to Table 1 there is a huge P gradient across your sites, therefore please discuss how this could possible affect your results.

Line 477 It is unclear which analysis this sentence is referring to. Your analysis or the work by Quesada et al 2012 mentioned in the previous sentence? Please clarify and elaborate on this statement.

Line 515 What do you mean by GPP is demand limited across spatial scale? Is it the atmospheric transpirational demand? And how does this relate to your already conducted analysis of you results? According to your own analysis, LAI is explaining most of the variation across the spatial scale (sec. 3.5.1, sec. 3.5.2, and 4.2). You must elaborate on the statement.

Line 517 In this section, or possible add another section to the discussion, you will also have to address the uncertainties from the intrinsic model behaviour. You recognised already in line 245 that the model uncertainty is underestimated due to the lack of intrinsic model error. However, during your discussion this is not assessed at all. Please assess these uncertainties in particular in relation to the moisture stress and how the plant hydraulics is modelled in SPA and acknowledge its limitations.

Line 544-546 Something is wrong with this part of sentences. Please rephrase.

[Figure]

Line 549-551 I agree. And to my knowledge, some models already do so.

Line 576 According to the manuscript preparation guidelines for authors for Biogeoscience, Journal names are to be abbreviated according to the Journal Title Abbreviations by Caltech Library
(see https://www.biogeosciences.net/for_authors/manuscript_preparation.html ).

Line 1010 Table 2, delete 'subsequently allowed to fluctuate'. This is already implied by being initial conditions.

Line 1095 Figure 1, $NPP_{Leaf}$ does not classify as a trait. Please correct the figure accordingly.

Supplement Material:

Please use accurate mathematical expression (e.g. then; should be $\Rightarrow$) and make sure the equations are not in italic.

Please consider numerating your equations

If the LAI is forced using monthly time series, then how does the LAI change between the daily time steps in the calculations for $NPP_{Leafi}$? Is it because you nudge the LAI, and hence not force it at every time step? Please clarify.

As mentioned already, the three latter equations are confusing. If the labile pool is depleted ($NSC_i < NSC_{i-1}$) then you allocate from the total NPP pool to the labile pool. I assume that this is only the case when $NPP_{Leafi}$ is met by the daily assimilation? Please clarify and complete your sets of equations for all cases.

**Technical corrections:**

In general, Please reconsider the usage of the word whilst – it reads a little pretentious.

Line 32 Abbreviate GPP in line 32, not line 34

Line 88 Abbreviate LAI in line 88, not line 90

Line 100 Please change p50 to $\psi$50

Line 195 Please Abbreviate NPP here

Line 307 Add a space between the two sentences.

Line 473-774 Add *they* to the sentence: . . . so they may be acclimated. . .

Line 716 Please change CO2 to $CO_2$

Line 716 Please change CO 2 to $CO_2$

Line 929 Please change CO 2 to $CO_2$

Line 981 Please change CO2 to $CO_2$

Line 996-999 Table 1: Please include the abbreviation of RAINFOR in the Table 1 text. Should RAINFOR be above the second horizontal line? Is there no site code for Tanguro? Please use correct degree symbol for the unit of Mean annual air temperature. Would it be possible to add species composition or just dominant species at each site to the table?

Line 1095 Please delete the single parenthesis in this line.

Line 1112 Please correct nodule to module.

---

## Referee Comment (RC2) · Anonymous Referee #2 · 6 Jul 2019

Review of Leaf area indexnchanges explain GPP variagion across an Amazon droughg stress bradient by Flank-Prain et al, Biogeosciences, 2019. This paper describes how GPP changes along a gradient are explained by direct and indirect effects of climate forcing in Amazonia. The direct forcings include physiological responses while indirect responses include ecosystem structural and leaf trait responses. The authors use observations and a calibrated soil, plant , atmosphere model to single out the different responses. They find that indirect reponses dominate the explanation of the spatial variation, whereas seasonal variation was dominated by response to global radiation, the strength of which depends on the level of drought. The paper is in general well written, well structured, concise and to the point. There are a few point that need attention: - In a few instances, which I will describe below, I found the formulation of

[Figure]

sentences vague. - In my opinion research question 3 adds little value to the paper and the corresponding results are relatively shallow relative to the existing literature. The results are quite obvious. I suggest removing this rq and the corresponding results. It will make the paper sharper and more to the point. - I miss a discussion on the temporal scale of the responses. The authors use 'multiple' (2) years of forcing data. They find that indirect responses dominate. I understand that the paper describes equilibrium responses to an existing drought stress gradient. Still indirect responses probably need some time to develop, while droughts are often intermittent. If I do not fully understand how the authors see this, it may indicate the need to discuss this issue explicitly. - The model methodology is explained insufficiently to allow for independent reproducibility of the results and for understanding what the authors really did. Specific comments - Line 90. Please explain on what time scale this evidence is valid. - Line 90.This paragraph is rather qualitative, therefore vague. Please explain how strong the responses are. - Line 104: understanding is limited. This is quite zn empty sentence. Please make it more concrete by stating what understanding is missing exactly. - Line 127. We link . . . . Vague sentence. Additionally, 2 years is really the minimal number of multiple years. Couldn't you use a longer data set? This is relevant to how fast ecosystems respond to and recover from drought. How do you capture transient responses and how do you know those 2 years are representative for average (or not extreme) conditions? - Line 171. Please define MCWD precisely. - Line 208. How frequent were data gaps? - Line 222. . . .overestimation. . . please quantify. - Line 254. This sentence confused me initially, it sounds like you are only focussed on direct effects. Please rephrase. - Line 359. Please clarify how this can be seen in table 3. - Section 3.5.2. This section is difficult to read because of the many numbers. Is presenting them in a table of figure an option? After these points have been addressed, I advise positively to publish this paper in Biogeosciences as it is a valuable contribution to better understanding spatial and temporal variations in GPP in Amazonia and the carbon cycle in general.

---

## Author Comment (AC1) · 22 Jul 2019

We thank the reviewer for their time and effort. We found their comments very helpful and considered all of them in the revised version of our manuscript.

Author responses to reviewer comments are denoted by AR.

General Comments The authors investigate the importance of different drivers (LAI, leaf traits, climate) for GPP at both temporal and spatial scale across a drought stress gradient in the amazonian region using the Soil-Plant-Atmosphere (SPA) ecosystem model. The SPA model is applied at 7 sites, using sites specific parameters and is forced with LAI observed from hemispherical photographs. Simulation experiments and machine learning techniques are used to investigate their scientific questions.

[Figure]

They find that indirect effects via plant traits and ecosystem structural changes, here expressed as LAI, are found to be the main driver of GPP across a spatial drought gradient, but the sensitivity of GPP to changes in these drivers varied with the gradient. On a sub-annual timescale climatic drivers were found to be more important for GPP. The authors discuss how these direct physiological and indirect mechanism affect GPP but fail to explain the added value of forcing their model with observed LAI and to explain in detail how this forced LAI propagates down the modelling structure of SPA. The manuscript is well written and well structured, however, with many repetitions that should be deleted to make space for more details on your methods. As explained in detail in the comments below, I would like the authors to consider my questions and comments, before I recommend the publication of this manuscript.

AR 1. The brief summary of how LAI is forced within the model (lines 232-237) has been expanded to provide a more in-depth description, and instead included in the '2.1 The Soil Plant Atmosphere model (SPA)' subsection (see comments AR 10 and AR 5). In addition, as suggested, we outline the value of using LAI data to force the model. Constraining simulated LAI using field observations was integral to the model experiments conducted. It allowed us to quantify the direct effects of different LAI timeseries on GPP under different plot conditions. If the LAI timeseries were not constrained we would be unable to quantify these effects.

Throughout the manuscripts, your hypothesis/conclusions are repeated (abstract, introduction, discussion, and conclusion). This takes up a lot of space that could otherwise have been used elsewhere in the manuscript. Therefore, I urge you to delete several of these repeated paragraphs. Please see the specific comments below for my suggestions.

AR 2. The repetition of hypotheses and conclusions has been revised as per the specific comments (AR 7, AR 8, AR 13 and AR 17).

The title Leaf Area Index Changes Explain GPP variations across an Amazon Drought

[Figure]

Stress Gradient is not surprising as LAI generally scales well with GPP, and hence you would expect the changes to do so as well. Moreover, as also stated in the manuscript, the changes of LAI are affected by drought stress, and thus it is indirectly the drought stress that is causing the variation in GPP. Lastly, the title does not fully cover all three research questions made by the authors in the manuscript, although it points towards your most interesting finding. However, I would suggest that you reconsider the title.

AR 3. The authors put forward an alternative title 'The importance of physiological, structural and trait responses to drought stress in driving spatial and temporal variation in GPP across Amazon forests'. Or simplified to 'The mechanisms driving spatial and temporal variation in GPP across an Amazon drought stress gradient'.

Several times you state that changes in LAI is an indirect structural effect from changes in soil moisture. From there, it follows that it is LAI which drives the GPP across the MCWD gradient. A strong emphasis is throughout the manuscript put on LAI and LAI as a driver of GPP, while LAI is strongly impacted by drought stress. However, the model is forced with LAI from hemispherical photographs, but the authors do not explain how the forced LAI is linked to and impact the simulated soil moisture content. From Fig. 2 it follows that LAI impacts the foliage carbon pool, and this pool together with carbon pool of fine roots and soil moisture impacts GPP, but the link between the forced LAI and soil moisture is not well explained for your model setup. Please clarify this in the manuscript.

AR 4. In the model, hydraulic transport is coupled across the soil-plant atmosphere continuum. As a result, higher evaporative demand under increased LAI drives increased root water uptake and consequently a depletion in soil moisture. The link between LAI and soil moisture within the model is now described in the '2.1 The Soil Plant Atmosphere model (SPA)' subsection.

Several times you briefly mention the C allocation (line 232 to 241, Fig. 2 and Supplement material). In the text you state that allocation to NPPleaf occurs first. Normally

NPP is considered a flux, and normally you would allocate to a pool. Thus, do you mean that allocation to the foliar stock occurs first? If assimilation does not provide the C need for allocation to support the LAI, you take from the labile/non-structural carbon pool. However, in the supplement material in the last three equations, you state that if the labile pool has been depleted you allocate from the total NPP. Surely this must only be the case when you have enough NPP to sustain the foliar stock as required by the LAI. Please clarify this in the manuscript.

AR 5. The reviewer is correct in their summation of the C allocation scheme. Allocation towards NSC storage is executed in subsequent time steps when the NPPleaf requirement does not exceed total NPP. This is now clarified in the manuscript.

Specific Comments

Line 88 can be read as if you say LAI is a trait. Conventionally, LAI is not considered a trait (you could use max LAI), but rather relates to the ecosystem structure. Thus, for clarification could you please consider rephrasing the sentence to e.g.: Changes to both ecosystem structure and traits, such as LAI, rooting depth and carboxylation capacity, are expected to be more longstanding (Meir et al., 2015a).

AR 6. Corrected as suggested

Line 142-162 As the introduction is already very long, and much of your hypothesis is repeated later in the manuscript, I would highly recommend deleting these paragraphs.

AR 7. See AR 8 comment

Line 175-181 These lines are almost identical to your scientific research questions listed in the introduction. Please consider deleting one or the other.

AR 8. Removed (175-181) as suggested

Line 232-234 You state that the mapping of canopy dynamics is critical, and that changes in canopy dynamics cause disparity between field observations and model

predictions – how well is canopy dynamics simulated by SPA? How is the LAI forced over the canopy layers in SPA? Please elaborate on these aspects in the manuscript and explain how your study improve these shortcomings.

AR 9. The capacity of SPA to accurately simulate canopy dynamics is demonstrated in López‐Blanco et al. (2018) Sus et al. (2010). However, simulated canopy dynamics in the presented manuscript are tightly constrained to observations (LAI and leaf litterfall are forced and calibrated using field measurements respectively). The vertical distribution of leaf area is kept constant throughout model experiments, as current field data is insufficient to provide an accurate depiction of how vertical distributions change with canopy density across the MCWD gradient. As suggested by the reviewer, we have expanded on these aspects and considered how this study could prove useful to model development. The presented study directs model development attentions towards the indirect-structural and trait effects of drought stress. The importance of canopy dynamics in driving GPP, suggests C allocation and plant traits including leaf lifespan will prove important to capturing climate change responses. Though the link between traits and canopy dynamics was outside the scope of this manuscript, it is the subject of ongoing research by the authors.

Line 232-237 NPPLeaf was calculated as the difference between the foliar C stock of the previous time step and that which would equate to field measured LAI. The field measured LAI has a monthly resolution. In principle you would have foliar C stock that could change at every model time step. But if the foliar C stock already equates to the field measured LAI, because the resolution of the forced LAI is monthly, NPPLeaf would just be zero.

AR 10. Field measurements of LAI were interpolated to scale from monthly to daily estimates. This is now clarified in the manuscript.

Line 239 This sentence is not clear.

AR 11. Edited to explain additional functions of the NSC pool such as phloem transport

and osmoregulation.

Line 243 How was the SPA model calibration constraint by an upper and lower sample error of LAI? Please add clarification to the text.

AR 12. The model was forced using the observed LAI timeseries plus and minus the sample error for each plot. This is now clarified in the text.

Line 301-302 These lines are repetitions, and not needed. Please consider deleting.

AR 13. Deleted

Line 308 There is something wrong with the structure of this sentence.

AR 14. Reworked. "A positive, but non-significant, correlation existed between model and field estimates of seasonal soil moisture range (R2=0.35, p=0.21, RMSE=5%)."

Line 320-325 The correlation between GEM estimated and SPA simulation GPP are non-significant and moderate. As GPP is imperative for your analysis, have you considered the impact it might have on you results? Have you investigated how the GEM estimated GPP relates to the LAI form the hemispherical photographs?

AR 15. Discussion on the GEM-SPA GPP comparison and its potential impact is now included in the limitation sections. We recognise that the lack of significant correlation between SPA and GEM GPP estimates could impact the interpretation of our results. However, we highlight the difficulties of having a small sample size (Line 416) and that five of the seven plots were within the error estimates of field measurements. We also note that current GEM GPP estimates are based on sample error and do not account for assumptions used in scaling to derive GPP estimates. In response to the reviewers question, GPP estimates are not significantly correlated with LAI (R2=0.18, p=0.33), likely due to concurrent shift towards higher photosynthetic capacity at drier sites.

Line 326 Please explain why the GPPSPA variance is calculated under the LAI standard error.

[Figure]

AR 16. LAI upper and lower sample error were used to calculate an estimate of SPA uncertainty based on model input data. With regards to other model inputs, the availability of trait uncertainty estimates was variable and plot dependent, and there were no uncertainty estimates for hourly meteorological data. We were consequently limited to LAI estimates.

Line 401-410 This paragraph sounds like a conclusion, and since you have a Conclusion section, where this is also stated, I would suggest you delete this paragraph.

AR 17. Deleted

Line 437-449 You mention how changes in LAI is a response to precipitations regimes, and even call it a key response mechanism. Then, one could infer that it is just as much changes in precipitation that explains the changes in GPP as it is LAI. You state that changes in LAI is an indirect effect from climate — although the impact might be somewhat delayed, but did you investigate lagged correlations and variances between precipitation and GPP?

AR 18. With respect to annual timescales the field data used were across 2 years only, limiting the scope to test for lag effects. However, this could certainly prove an interesting investigation for the future. Across sub-annual timescales we are unable to compare field estimates of GPP (due to the nature of biometric estimates), however model experiment 3 found that the effect of moisture stress was better captured via VPD (line 392).

Line 452 You do not have a forest at steady state, if it is changing and experiencing trends. Please explain what you mean by a steady state forest – is it continuous cover, constant number of trees etc.? Or better yet refrain from using the term.

AR 19. Term omitted

Line 465-468 With the SPA model you are not able to quantify the impact of soil nutrient on the GPP-MCWD interactions because you lack nutrient cycling in the SPA model.

However, according to Table 1 there is a huge P gradient across your sites, therefore please discuss how this could possible affect your results.

AR 20. As the reviewer highlights soil nutrients varies widely across sites. However we note that there is no significant interaction between GPP and soil phosphorous (GEM $R^2$= 0.1, p=0.48; SPA $R^2$=0.01, p=0.81) or soil nitrogen (GEM $R^2$=0.37 , p=0.14; SPA $R^2$=0.31, p=0.19). We recognise that nutrient limitation likely impacts GPP across the MCWD gradient, effected through both nutrient availability and plants acquisition capacity (which is dependent on moisture-stress). We expect to capture soil nutrient effects in part via the inclusion of site specific leaf nutrient estimates as a model inputs (which influence simulated photosynthetic and metabolic rate). However, our assumption is limited by the divergence in canopy strategies across the gradient which likely impacts how plants use available nutrients. Discussion on nutrient limitation is now included in '4.5 Limitations and Opportunities'.

Line 477 It is unclear which analysis this sentence is referring to. Your analysis or the work by Quesada et al 2012 mentioned in the previous sentence? Please clarify and elaborate on this statement.

AR 21. This statement refers to the presented analysis (Experiment 1). Differences in root depth and biomass can alleviate water constraints to photosynthesis via the direct physiological pathway (i.e stomatal conductance). But in the absence of C cycle feedbacks, changes in root depth and biomass do not drive changes in emergent canopy properties (i.e. LAI) which proved most important in determining GPP. This is now clarified in the text.

Line 515 What do you mean by GPP is demand limited across spatial scale? Is it the atmospheric transpirational demand? And how does this relate to your already conducted analysis of you results? According to your own analysis, LAI is explaining most of the variation across the spatial scale (sec. 3.5.1, sec. 3.5.2, and 4.2). You must elaborate on the statement.

AR 22. This statement referred to the role of ecosystem structure and traits (i.e. demand) in determining spatial variation in GPP, in comparison to the supply limit observed across sub-annual timescales. However we recognise that the statement was unclear and have amended it to the following "Our results indicate that with respect to soil moisture, GPP is constrained via indirect pathways (i.e. ecosystem structure and traits) across spatial scales, but is limited via direct pathways (i.e. physiology) across sub-annual timescales (Figure 1)."

Line 517 In this section, or possible add another section to the discussion, you will also have to address the uncertainties from the intrinsic model behaviour. You recognised already in line 245 that the model uncertainty is underestimated due to the lack of intrinsic model error. However, during your discussion this is not assessed at all. Please assess these uncertainties in particular in relation to the moisture stress and how the plant hydraulics is modelled in SPA and acknowledge its limitations.

AR 23. As suggested, we have added a discussion on model limitations into the '4.5 Limitations and Opportunities' section. As evidenced by Bonan et al. (2014) the representation of plant hydraulics is a particular strength for SPA. However the model is not without its limitations. Notably, SPAs use of community weighted mean traits (and trait distributions) could influence the simulated response to environmental change.

Line 544-546 Something is wrong with this part of sentences. Please rephrase.

AR 24. Rephrased. "acquisition potential (Q2). We identify the potential vulnerability"

Line 549-551 I agree. And to my knowledge, some models already do so. Line 576 According to the manuscript preparation guidelines for authors for Biogeoscience, Journal names are to be abbreviated according to the Journal Title Abbreviations by Caltech Library (see https://www.biogeosciences.net/for_authors/manuscript_preparation.html ).

AR 25. Corrected

Line 1010 Table 2, delete 'subsequently allowed to fluctuate'. This is already implied by being initial conditions.

AR 26. Corrected

Line 1095 Figure 1, NPPLeaf does not classify as a trait. Please correct the figure accordingly.

AR 27. Corrected

Supplement Material:

Please use accurate mathematical expression (e.g. then; should be ⇒) and make sure the equations are not in italic.

AR 28. Corrected

Please consider numerating your equations If the LAI is forced using monthly time series, then how does the LAI change between the daily time steps in the calculations for NPPLeaf i? Is it because you nudge the LAI, and hence not force it at every time step? Please clarify.

AR 29. Numbering added. See AR 10 for clarification on LAI change at each timestep. As mentioned already, the three latter equations are confusing. If the labile pool is depleted ($NSC_i < NSC_{i-1}$) then you allocate from the total NPP pool to the labile pool. I assume that this is only the case when $NPPLeaf_i$ is met by the daily assimilation? Please clarify and complete your sets of equations for all cases

AR 30. See AR 5.

Technical corrections:

In general, Please reconsider the usage of the word whilst – it reads a little pretentious.

AR 31. Corrected

Line 32 Abbreviate GPP in line 32, not line 34

AR 32. Corrected

Line 88 Abbreviate LAI in line 88, not line 90

AR 33. Corrected

Line 100 Please change p50 to $\psi 50$

AR 34. Corrected

Line 195 Please Abbreviate NPP here

AR 35. Corrected

Line 307 Add a space between the two sentences.

AR 36. Corrected

Line 473-774 Add they to the sentence: . . . so they may be acclimated. . .

AR 37. Corrected

Line 716 Please change CO2 to CO2

AR 38. Corrected to subscript

Line 716 Please change CO 2 to CO2

AR 39. Corrected to subscript

Line 929 Please change CO 2 to CO2

AR 40. Corrected to subscript

Line 981 Please change CO2 to CO2

AR 41. Corrected to subscript

Line 996-999 Table 1: Please include the abbreviation of RAINFOR in the Table 1 text.

AR 42. Added

Should RAINFOR be above the second horizontal line?

AR 43. No, the row name should read 'RAINFOR site code'. This could be resolved by adjusting table dimensions or adding a hyphen.

Is there no site code for Tanguro?

AR 44. No, it is not on the RAINFOR database (though RAINFOR does hold data on other Tanguro plots).

Please use correct degree symbol for the unit of Mean annual air temperature.

AR 45. Corrected

Would it be possible to add species composition or just dominant species at each site to the table?

AR 46. This could be added. However, species richness varies between 65 and 195 across plots, with the most common species typically accounting for less than 20% of stems.

Line 1095 Please delete the single parenthesis in this line.

AR 47. Corrected

Line 1112 Please correct nodule to module.

AR 48. This was supposed to refer to 'nodule' however as this does not read well we have changed it to node.

References

BONAN, G. B., WILLIAMS, M., FISHER, R. A. & OLESON, K. W. 2014. Modeling stomatal conductance in the earth system: linking leaf water-use efficiency and water transport along the soil–plant–atmosphere continuum. Geoscientific Model Development, 7, 2193-2222. LÓPEZ‐BLANCO, E., LUND, M., CHRISTENSEN, T. R., TAMSTORF, M. P., SMALLMAN, T. L., SLEVIN, D., WESTERGAARD‐NIELSEN,
A., HANSEN, B. U., ABERMANN, J. & WILLIAMS, M. 2018. Plant traits are key determinants in buffering the meteorological sensitivity of net carbon exchanges of Arctic tundra. Journal of Geophysical Research: Biogeosciences, 123, 2675-2694. SUS, O., WILLIAMS, M., BERNHOFER, C., BÉZIAT, P., BUCHMANN, N., CESCHIA, E., DOHERTY, R., EUGSTER, W., GRÜNWALD, T. & KUTSCH, W. 2010. A linked carbon cycle and crop developmental model: Description and evaluation against measurements of carbon fluxes and carbon stocks at several European agricultural sites. Agriculture, ecosystems & environment, 139, 402-418.

---

## Author Comment (AC2) · 22 Jul 2019

We thank the reviewer for their time and effort and very helpful comments. Detailed answers to the comments are provided below. Author responses are denoted by AR.

Review of Leaf area index changes explain GPP variation across an Amazon drought stress gradient by Flack-Prain et al, Biogeosciences, 2019. This paper describes how GPP changes along a gradient are explained by direct and indirect effects of climate forcing in Amazonia. The direct forcings include physiological responses while indirect responses include ecosystem structural and leaf trait responses. The authors use observations and a calibrated soil, plant, atmosphere model to single out the different responses. They find that indirect responses dominate the explanation of the spatial

variation, whereas seasonal variation was dominated by response to global radiation, the strength of which depends on the level of drought. The paper is in general well written, well structured, concise and to the point. There are a few point that need attention:

- In a few instances, which I will describe below, I found the formulation of sentences vague.

AR 1. Edited in accordance with specific comments

- In my opinion research question 3 adds little value to the paper and the corresponding results are relatively shallow relative to the existing literature. The results are quite obvious. I suggest removing this rq and the corresponding results. It will make the paper sharper and more to the point.

AR 2. We take on board the reviewer's point with regards to RQ3. However, the reviewer's subsequent comment regarding discussion on the temporal scale of re- sponses prompted additions to the manuscript which we feel highlights the importance of RQ3s inclusion (see AR 3).

- I miss a discussion on the temporal scale of the responses. The authors use 'multiple' (2) years of forcing data. They find that indirect responses dominate. I understand that the paper describes equilibrium responses to an existing drought stress gradient. Still indirect responses probably need some time to develop, while droughts are often intermittent. If I do not fully understand how the authors see this, it may indicate the need to discuss this issue explicitly.

AR 3. We thank the reviewer for highlighting an important issue around the definition of drought stress. In the presented study we focused on seasonal drought stress, and compared GPP drivers across seasons and across a gradient in seasonal drought stress. In the original version of the manuscript a clear distinction between seasonal drought stress (our focus) and drought events (not addressed in the manuscript) was

not made. By defining drought stress in the context of our study early on we hope to ensure our references to drought stress are not ambiguous. The temporal scale of responses can then be discussed by comparing model experiments 1 and 3. We appreciate that the comparison was somewhat limited in the original version of the manuscript and as such have expanded the discussion (on from line 516).

- The model methodology is explained insufficiently to allow for independent reproducibility of the results and for understanding what the authors really did.

AR 4. A new figure has been added to the supplementary material detailing the inputs of each model run used in the analysis. We hope this figure helps sufficiently explain our methodology. However, we would be happy to consider further changes if the reviewer could provide more information on which details they believe to be missing.

Specific comments

- Line 90. Please explain on what time scale this evidence is valid.

AR 5. We have now included detail on the relevant timescale. Evidence exists across sub-annual (Araujo-Murakami et al., 2014, Xu et al., 2016) and annual timescales (Brando et al., 2008, Meir et al., 2009).

- Line 90. This paragraph is rather qualitative, therefore vague. Please explain how strong the responses are.

AR 6. Leaf area index (LAI) typically decreases with increasing drought stress (Meir et al., 2015a, Grier and Running, 1977). Across the wet-dry tropical forest transition LAI declines on average ∼1.4 m2m-2 (Iio et al., 2014). Temporal changes in drought stress report a 21-26% decline in LAI following five years of drought onset (via throughfall exclusion at Tapajós National Forest, Pará, Brazil; Brando et al., 2008). With respect to near surface root mass, length and surface area growth declines with seasonal water deficit, the paper referenced (Metcalfe et al., 2008) does not provide estimates on the strength of the response, only that is it significant (p<0.001). However, from the figure

presented we can estimate that root mass, length and surface area growth decline by up to 75%, 65% and 25% respectively (approximated using figure data retrieval software). Quantitative details have now been added to this section.

- Line 104: understanding is limited. This is quite an empty sentence. Please make it more concrete by stating what understanding is missing exactly.

AR 7. On reflection this sentence repeated (but with less specifics) the earlier statement of "The relative importance of plant physiology, ecosystem structure, and trait composition responses in determining variation in GPP, remain largely unquantified in data-constrained analysis (Meir et al., 2015b)." We have therefore removed it.

- Line 127. We link . . . . Vague sentence. Additionally, 2 years is really the minimal number of multiple years. Couldn't you use a longer data set? This is relevant to how fast ecosystems respond to and recover from drought. How do you capture transient responses and how do you know those 2 years are representative for average (or not extreme) conditions?

AR 8. With respect to the reviewers comment on drought response and recovery please see AR 3. In response to the reviewers question about using a longer data set. We were limited by the length of the timeseries available across plots of different data streams and the need to keep the number of annual GPP estimates from each plot consistent for a balanced statistical analysis.

- Line 171. Please define MCWD precisely.

AR 9. The introductory methods section has been edited as follows, and the MCWD equation added to the supplementary material. MCWD is the maximum cumulative water deficit reached within a year. A water deficit estimate for each month is calculated as the difference between precipitation and transpiration (which ground measurements estimate at âĹij100 mm month−1, see Aragao et al. (2007)). Therefore, the forest is in water deficit if monthly precipitation falls below 100mm. Maximum cumulative water

deficit is calculated as the sum of sequential monthly water deficits (see supplementary material for equation).

- Line 208. How frequent were data gaps?

AR 10. The number of missing hourly field meteorological measurements across the timeseries varied from 2-40% across sites, whilst the frequency of gaps varied from 2-99 yr-1. Gaps less than 6 hours in length accounted for between 20-100% of total gaps across plots. Statistics on the frequency of data gaps has now been added to the manuscript.

- Line 222. . . .overestimation. . . please quantify.

AR 11. Pre-calibration, SPA soil moisture estimates were on average 11-68% higher than field measurements across plots. The difference between model and field soil moisture estimates increased significantly with MCWD (R2=0.69, p=0.04).

- Line 254. This sentence confused me initially, it sounds like you are only focussed on direct effects. Please rephrase.

AR 12. Rephrased to "Our aim was to isolate the direct effects of climate and soils (via physiological responses), and the indirect effects via ecosystem structure, and leaf traits, on simulated GPP." - Line 359. Please clarify how this can be seen in table 3.

AR 13. This was an error and should have referenced the preceding section ('3.4 LAI and Leaf Trait Trends along the MCWD gradient'). Now corrected.

- Section 3.5.2. This section is difficult to read because of the many numbers. Is presenting them in a table of figure an option?

AR 14. The authors have moved the site specific Δ values to Figure 6.

After these points have been addressed, I advise positively to publish this paper in Biogeosciences as it is a valuable contribution to better understanding spatial and temporal variations in GPP in Amazonia and the carbon cycle in general.

[Figure]

References ARAGAO, L. E. O. C., MALHI, Y., ROMAN-CUESTA, R. M., SAATCHI, S., ANDERSON, L. O. & SHIMABUKURO, Y. E. 2007. Spatial patterns and fire response of recent Amazonian droughts. Geophysical Research Letters, 34. ARAUJO-MURAKAMI, A., DOUGHTY, C. E., METCALFE, D. B., SILVA-ESPEJO, J. E., ARROYO, L., HEREDIA, J. P., FLORES, M., SIBLER, R., MENDIZABAL, L. M., PARDO-TOLEDO, E., VEGA, M., MORENO, L., ROJAS-LANDIVAR, V. D., HALLADAY, K., GIRARDIN, C. A. J., KILLEEN, T. J. & MALHI, Y. 2014. The productivity, allocation and cycling of carbon in forests at the dry margin of the Amazon forest in Bolivia. Plant Ecology & Diversity, 7, 55-69. BRANDO, P. M., NEPSTAD, D. C., DAVIDSON, E. A., TRUMBORE, S. E., RAY, D. & CAMARGO, P. 2008. Drought effects on litterfall, wood production and belowground carbon cycling in an Amazon forest: results of a throughfall reduction experiment. Philos Trans R Soc Lond B Biol Sci, 363, 1839-48. GRIER, C. C. & RUNNING, S. W. 1977. Leaf Area of Mature Northwestern Coniferous Forests - Relation to Site Water-Balance. Ecology, 58, 893-899. IIO, A., HIKOSAKA, K., ANTEN, N. P. R., NAKAGAWA, Y. & ITO, A. 2014. Global dependence of field‐observed leaf area index in woody species on climate: a systematic review. Global Ecology and Biogeography, 23, 274-285. MEIR, P., BRANDO, P. M., NEPSTAD, D., VASCONCELOS, S., COSTA, A. C. L., DAVIDSON, E., ALMEIDA, S., FISHER, R. A., SOTTA, E. D. & ZARIN, D. 2009. The effects of drought on Amazonian rain forests. Amazonia and Global Change, 429-449. MEIR, P., MENCUCCINI, M. & DEWAR, R. C. 2015a. Drought-related tree mortality: addressing the gaps in understanding and prediction. New Phytol, 207, 28-33. MEIR, P., WOOD, T. E., GALBRAITH, D. R., BRANDO, P. M., DA COSTA, A. C., ROWLAND, L. & FERREIRA, L. V. 2015b. Threshold Responses to Soil Moisture Deficit by Trees and Soil in Tropical Rain Forests: Insights from Field Experiments. Bioscience, 65, 882-892. METCALFE, D. B., MEIR, P., ARAGAO, L. E. O. C., DA COSTA, A. C. L., BRAGA, A. P., GONCALVES, P. H. L., SILVA, J. D., DE ALMEIDA, S. S., DAWSON, L. A., MALHI, Y. & WILLIAMS, M. 2008. The effects of water availability on root growth and morphology in an Amazon rainforest. Plant and Soil, 311, 189-199. XU, X., MEDVIGY, D., POWERS, J. S., BECKNELL, J. M. & GUAN,

[Figure]

K. 2016. Diversity in plant hydraulic traits explains seasonal and inter‐annual variations of vegetation dynamics in seasonally dry tropical forests. New Phytologist, 212, 80-95.

---

## Author Response (AR1)

**The Importance of Physiological, Structural and Trait Responses to Drought Stress in Driving Spatial and Temporal Variation in GPP across Amazon Forests**

Sophie Flack-Prain[1], Patrick Meir[1,2], Yadvinder Malhi[4], Thomas Luke Smallman[1,3], Mathew Williams[1,3]

[1] School of GeoSciences, University of Edinburgh, Edinburgh, UK

[2] Research School of Biology, Australian National University, Canberra, ACT, Australia

[3] National Centre for Earth Observation, University of Edinburgh, UK

[4] Environmental Change Institute, School of Geography and the Environment, University of Oxford, Oxford, UK

*Correspondence to:* Sophie Flack-Prain (s.flack-prain@ed.ac.uk)

> **Commented [FS1]:** R1 AR3. In response to the reviewers comment "The title Leaf Area Index Changes Explain GPP variations across an Amazon Drought Stress Gradient is not surprising as LAI generally scales well with GPP, and hence you would expect the changes to do so as well. Moreover, as also stated in the manuscript, the changes of LAI are affected by drought stress, and thus it is indirectly the drought stress that is causing the variation in GPP. Lastly, the title does not fully cover all three research questions made by the authors in the manuscript, although it points towards your most interesting finding. However, I would suggest that you reconsider the title."
>
> The authors put forward an alternative title 'The importance of physiological, structural and trait responses to drought stress in driving spatial and temporal variation in GPP across Amazon forests'. Or simplified to 'The mechanisms driving spatial and temporal variation in GPP across an Amazon drought stress gradient'.

**Abstract**

The capacity of Amazon forests to sequester carbon is threatened by climate change-induced shifts in precipitation patterns. However, the relative importance of plant physiology, ecosystem structure, and trait composition responses in determining variation in gross primary productivity (GPP), remain largely unquantified, and vary among models. We evaluate the relative importance of key climate constraints to  (GPP), comparing direct plant physiological responses to water availability and indirect structural and trait responses (via changes to leaf area index (LAI), roots and photosynthetic capacity). To separate these factors we combined the Soil-Plant-Atmosphere model with forcing and observational data from seven intensively studied forest plots along an Amazon drought stress gradient. We also used machine learning to evaluate the relative importance of individual climate factors across sites. Our model experiments showed that variation in LAI was the principal driver of differences in GPP across the gradient, accounting for 33% of observed variation. Differences in photosynthetic capacity ($V_{cmax}$ and $J_{max}$) accounted for 21% of variance, and climate (which included physiological responses) accounted for 16%. Sensitivity to differences in climate was highest where shallow rooting depth was coupled with high LAI. On sub-annual timescales, the relative importance of LAI in driving GPP increased with drought stress ($R^2=0.72$), coincident with decreased  importance of solar radiation  ($R^2=0.90$). Given the role of LAI in driving GPP across Amazon forests, improved mapping of canopy dynamics is critical, opportunities for which are offered by new satellite-based remote sensing missions such as GEDI, Sentinel and FLEX.

Keywords: Canopy Dynamics, Leaf Traits, Tropical Rainforests, Precipitation, Gross Primary Productivity

**Commented [FS2]:** R1 AR32. in response to reviewer comment "Line 32 Abbreviate GPP in line 32, not line 34"

**Commented [FS3]:** R1 AR31. In response to reviewer comment "In general, Please reconsider the usage of the word whilst – it reads a little pretentious." Adjustments throughout

**1. Introduction**

As the entry point for carbon into the biosphere, gross primary productivity (GPP) is central to the

60 global carbon cycle. Tropical rainforests alone account for one third of total terrestrial  GPP, assimilating ~41 Pg of carbon each year (Beer et al., 2010). Carbon fluxes across the tropics are tightly coupled to climate, and water availability is a principal driver of spatial and temporal variation in  GPP (Fisher et al., 2007, Von Randow et al., 2013, Beer et al., 2010, Malhi et al., 2015, Guan et al., 2015). Across Amazon forests, GPP decreases linearly with increasing seasonal water

65 deficit (Malhi et al., 2015). Shifts in precipitation patterns as a result of anthropogenic climate change are predicted to have a major impact on Amazon GPP (Phillips et al., 2009, Malhi et al., 2008, Meir and Woodward, 2010, Zhang et al., 2015, Meir et al., 2015a). Longer and more intense dry seasons are projected, together with an increased frequency and severity of drought events (Joetzjer et al., 2013, Boisier et al., 2015, Duffy et al., 2015). Given the biogeochemical influence of Amazon forests at

70 regional and global scales (Liu et al., 2017), accurately predicting GPP response to drought stress is critical.

Dynamic global vegetation models (DGVMs) disagree on the effects of projected precipitation change on Amazon carbon dynamics. Galbraith et al. (2010) found future shifts in precipitation patterns had little effect on model estimates of biomass change (for two of the three models tested), reflecting poorly

75 the observed sensitivity of Amazon forests to water availability illustrated by through-fall exclusion experiments and natural drought events (Rowland et al., 2015a, Nepstad et al., 2007, Phillips et al., 2009). Substantial progress has been made in model development to capture the impact of drought stress on plant physiology. By coupling stomatal conductance and plant hydraulic theory, models have proved better able to predict ecosystem functioning and mortality (Eller et al., 2018, Fisher et al., 2018, Fisher

80 et al., 2006, Fisher et al., 2007, Bonan et al., 2014). However, the interactions between drought stress, ecosystem structure (e.g. canopy dynamics and rooting depth) and trait composition (e.g. $V_{cmax}$, $J_{max}$, leaf lifespan and leaf mass per unit area (LMA)), are typically absent from models, despite having a

**Commented [FS4]:** R2 AR1. In response to the reviewers comment "- In a few instances, which I will describe below, I found the formulation of sentences vague."
Edited in accordance with specific comments

**Commented [FS5]:** R1 AR2. In response to the reviewers comment "Throughout the manuscripts, your hypothesis/conclusions are repeated (abstract, introduction, discussion, and conclusion). This takes up a lot of space that could otherwise have been used elsewhere in the manuscript. Therefore, I urge you to delete several of these repeated paragraphs. Please see the specific comments below for my suggestions."

The repetition of hypotheses and conclusions has been revised as per the specific comments (see AR7, AR8, AR13 and AR17).

major impact on simulated GPP (Fauset et al., 2012, Sakschewski et al., 2016, Lee et al., 2013).
Furthermore, changes in canopy dynamics have been identified as a likely cause for the disparity
between field observations and model predictions (Restrepo-Coupe et al., 2017, Powell et al., 2013).

The relative importance of plant physiology, ecosystem structure, and trait composition responses in determining variation in GPP, remain largely unquantified in data-constrained analysis (Meir et al., 2015b). Plant physiological responses to drought stress include stomatal conductance, which is limited by water availability and atmospheric demand. Stomatal conductance constrains GPP via changes in $CO_2$ supply, but is considered a short (varying on sub-hourly timescales), rather than long-term response to climate forcings (Sperry et al., 2002). Changes to both ecosystem structure and traits, such as leaf area index (LAI), rooting depth and carboxylation capacity, are expected to be more longstanding (Meir et al., 2015a).

Extensive evidence links spatial and temporal variation in drought stress with ecosystem structure (across sub-annual and annual timescales). LAI typically decreases with increasing drought stress (Iio et al., 2014, Meir et al., 2015b, Brando et al., 2008, Grier and Running, 1977, Wright et al., 2013).  Across the wet-dry tropical forest transition, LAI declines on average ~1.4 $m^2m^{-2}$ (Iio et al., 2014). Brando et al. (2008) report a 21-26% decline in LAI following five years of drought onset at the Amazon throughfall exclusion experiment at Tapajós National Forest, Pará, Brazil. Growth of near surface root mass, length and surface area decline with seasonal drought stress (and increase during periods of high soil water availability to exploit available resources), whilst deep roots can support water supply during dry periods (Nepstad et al., 1994, Metcalfe et al., 2008). Root depth, mass and traits influence hydraulic supply and consequently stomatal conductance.

Leaf traits similarly exhibit spatial and temporal variation with changing water availability. Leaf nitrogen content (per unit mass), light- and $CO_2$-saturated photosynthetic rates (per unit mass) increase with drought stress across tropical precipitation gradients,  $\psi$50 (the water potential at which 50% of hydraulic conductivity is lost) declines (Wright et al., 2004, Santiago et al., 2004, Anderegg,

Commented [FS6]: R1 AR6. In response to reviewer comment "Line 88 can be read as if you say LAI is a trait. Conventionally, LAI is not considered a trait (you could use max LAI), but rather relates to the ecosystem structure. Thus, for clarification could you please consider rephrasing the sentence to e.g.: Changes to both ecosystem structure and traits, such as LAI, rooting depth and carboxylation capacity, are expected to be more longstanding (Meir et al., 2015a)."
Corrected as suggested

Commented [FS7]: R2 AR5. In response to reviewer comment "Line 90. Please explain on what time scale this evidence is valid."
We have now included detail on the relevant timescale. Evidence exists across sub-annual (Araujo-Murakami et al., 2014, Xu et al., 2016) and annual timescales (Brando et al., 2008, Meir et al., 2009).

Commented [FS8]: R1 AR33. In response to reviewer comment "Line 88 Abbreviate LAI in line 88, not line 90"

Commented [FS9]: R2 AR6. In response to reviewer comment "Line 90. This paragraph is rather qualitative, therefore vague. Please explain how strong the responses are."

Quantitative details have now been added to this section.

With respect to near surface root mass, length and surface area growth declines with seasonal water deficit, the paper referenced (Metcalfe et al., 2008) does not provide estimates on the strength of the response, only that is it significant (p<0.001). However, from the figure presented we can estimate that root mass, length and surface area growth decline by up to 75%, 65% and 25% respectively (approximated using figure data retrieval software)."

Commented [FS10]: R1 AR31. In response to reviewer comment "In general, Please reconsider the usage of the word whilst – it reads a little pretentious."
Adjustments throughout

Commented [FS11]: R1 AR34. In response to reviewer comment "Line 100 Please change p50 to ψ50"
Corrected

110 2015). Leaf traits affect GPP via photosynthetic capacity ($V_{cmax}$ and $J_{max}$) (Bahar et al., 2017, Fyllas et al., 2017), and through their influence on canopy carbon economics, via leaf growth and maintenance costs (Bloom et al., 1985).

Field observations show variation in Amazon GPP is correlated with physiological, ecosystem structure
115 and trait composition responses to climate (Restrepo-Coupe et al., 2013, Goulden et al., 2004, Hutyra et al., 2007, Wu et al., 2017, Wagner et al., 2017). Modelling approaches have similarly highlighted the role of canopy dynamics and leaf traits in driving spatial and temporal variation in GPP (Mercado et al., 2011, Castanho et al., 2013, Restrepo-Coupe et al., 2013, Rodig et al., 2018), however their relative effects have not been explicitly isolated and quantified. Quantifying the direct effect of discrete
120 photosynthetic drivers has been limited by the need for detailed field measurements of carbon fluxes, canopy dynamics and traits. a deserved research effort has focused on the importance of nutrient availability in driving spatial variation in GPP (Mercado et al., 2011, Castanho et al., 2013), _however_ the role of ecosystem responses to water availability has received less attention (Green et al., 2019). In light of projected changes in rainfall patterns across the basin, capturing
125 responses to water availability in ecosystem models is critical to reducing current uncertainty around Amazon climate-vegetation feedbacks. We aim to reduce the uncertainty by assessing the relative effects of physiological, structural and trait responses to water availability on GPP across monthly to annual timescales.

We apply _an ecosystem_ model to plots across the Amazon, spanning a large
130 drought stress gradient _(herein, the term drought stress refers to seasonal water deficit)_, and a range in forest types from moist equatorial to seasonally dry tropical forests. Process modelling allows the links between climate, ecosystem structure and leaf traits to be quantified explicitly, and separated, across timescales (Figure 1). The soil plant atmosphere model (SPA) (Williams et al., 1996, Williams et al., 1998, Fisher et al., 2006, Fisher et al., 2007, Rowland et al., 2015b) is well suited to this investigation
135 given its prior use in accurately simulating carbon and water fluxes in Amazon tropical forests. We calibrate and validate the model using field data gathered over multiple years (2009-

Commented [FS12]: R2 AR7. In response to the reviewer comment "Line 104: understanding is limited. This is quite an empty sentence. Please make it more concrete by stating what understanding is missing exactly."
On reflection this sentence repeated (but with less specifics) the earlier statement of "The relative importance of plant physiology, ecosystem structure, and trait composition responses in determining variation in GPP, remain largely unquantified in data-constrained analysis (Meir et al., 2015b)." We have therefore removed it.

Commented [FS13]: R1 AR31. In response to reviewer comment "In general, Please reconsider the usage of the word whilst – it reads a little pretentious."
Adjustments throughout

Commented [FS14]: R2 AR3. In response to the reviewer comment "- I miss a discussion on the temporal scale of the responses. The authors use 'multiple' (2) years of forcing data. They find that indirect responses dominate. I understand that the paper describes equilibrium responses to an existing drought stress gradient. Still indirect responses probably need some time to develop, while droughts are often intermittent. If I do not fully understand how the authors see this, it may indicate the need to discuss this issue explicitly."
We thank the reviewer for highlighting an important issue around the definition of drought stress. In the presented study we focused on seasonal drought stress, and compared GPP drivers across seasons and across a gradient in seasonal drought stress. In the original version of the manuscript a clear distinction between seasonal drought stress (our focus) and drought events (not addressed in the manuscript) was not made. By defining drought stress in the context of our study early on we hope to ensure our references to drought stress are not ambiguous. The temporal scale of responses can then be discussed by comparing model experiments 1 and 3. We appreciate that the comparison was somewhat limited in the original version of the manuscript and as such have expanded the discussion (on from line 516).

2010) on permanent sample plots from the Global Ecosystems Monitoring (GEM) network (Doughty et al., 2015a, Malhi et al., 2015). The datasets comprise detailed measurements of carbon fluxes, carbon stocks and leaf traits, and were used to constrain the SPA model. We simulate the effect of forest structure and leaf trait distributions along the drought stress gradient, and explore the covariation of observed leaf traits (leaf N content (a proxy for photosynthetic capacity) and LMA) and those derived from model calibrations (leaf lifespan), before using SPA to address the following questions:

1. Is spatial variation in GPP across the drought stress gradient principally driven by the direct effects of climate and soils, which include physiological responses to water availability via hydraulic transport and stomatal conductance? Alternatively, are indirect effects of climate, via structural and trait responses to water availability (LAI, rooting biomass, root depth and photosynthetic capacity i.e $V_{cmax}$ and $J_{max}$), more important?

2. Does the sensitivity of GPP to differences in climate, LAI, photosynthetic capacity ($V_{cmax}$ and $J_{max}$) and rooting depth vary across the drought stress gradient?

3. What drives seasonal variation in GPP across an Amazon forest drought stress gradient?

Linked to question one, we hypothesise that indirect effects of climate via structural and trait responses are more important than the direct effects (via physiological responses), in explaining spatial variation in GPP across the drought stress gradient (Figure 1). We further posit that LAI is the principal driver of differences in GPP among Amazon forests, effected through the observed increase in leaf area with decreasing drought stress.

For question two, we predict that the sensitivity of GPP to differences in climate, LAI, photosynthetic capacity ($V_{cmax}$ and $J_{max}$) and rooting depth will vary dependent on water demand (via LAI and stomatal conductance) and supply (climate and root depth and biomass; Figure 1). We expect that forests under lower drought stress will be most sensitive to differences in LAI and photosynthetic capacity within the bounds of observations across the gradientthe gradient. We predict that forests under higher drought stress will be more sensitive to differences in rooting depth. We expect forests with high LAI but shallow rooting depth will be most sensitive to differences in climate, due to their higher transpiration demand but low capacity for water supply.

Commented [FS15]: R2 AR8. In response to reviewer comment "Line 127. We link . . . . Vague sentence. Additionally, 2 years is really the minimal number of multiple years. Couldn't you use a longer data set? This is relevant to how fast ecosystems respond to and recover from drought. How do you capture transient responses and how do you know those 2 years are representative for average (or not extreme) conditions?"
Sentence edited.
With respect to the reviewers comment on drought response and recovery please see AR 3. In response to the reviewers question about using a longer data set. We were limited by the length of the timeseries available across plots of different data streams.

Commented [FS16]: R2 AR2. In response to the reviewers comment "- In my opinion research question 3 adds little value to the paper and the corresponding results are relatively shallow relative to the existing literature. The results are quite obvious. I suggest removing this rq and the corresponding results. It will make the paper sharper and more to the point."
We take on board the reviewer's point with regards to RQ3. However, the reviewer's subsequent comment regarding discussion on the temporal scale of responses prompted additions to the manuscript which we feel highlights the importance of RQ3s inclusion (see AR 3).

[revised manuscript text omitted]

Commented [FS18]: R2 AR9. In response to reviewer comment "- Line 171. Please define MCWD precisely." Also see supplementary material.

Commented [FS19]: R1 AR7&8. In response to reviewer comments "Line 142-162 As the introduction is already very long, and much of your hypothesis is repeated later in the manuscript, I would highly recommend deleting these paragraphs."
And
"Line 175-181 These lines are almost identical to your scientific research questions listed in the introduction. Please consider deleting one or the other."
Removed as suggested

Commented [FS20]: R1 AR4. In response to the reviewer comment "Several times you state that changes in LAI is an indirect structural effect from changes in soil moisture. From there, it follows that it is LAI which drives the GPP across the MCWD gradient. A strong emphasis is throughout the manuscript put on LAI and LAI as a driver of GPP, while LAI is strongly impacted by drought stress. However, the model is forced with LAI from hemispherical photographs, but the authors do not explain how the forced LAI is linked to and impact the simulated soil moisture content. From Fig. 2 it follows that LAI impacts the foliage carbon pool, and this pool together with carbon pool of fine roots and soil moisture impacts GPP, but the link between the forced LAI and soil moisture is not well explained for your model setup. Please clarify this in the manuscript."

The link between LAI and soil moisture within the model is now described in the '2.1 The Soil Plant Atmosphere model (SPA)' subsection.

Commented [FS21]: R1 AR9. In response to the reviewer comment "Line 232-234 You state that the mapping of canopy dynamics is critical, and that changes in canopy dynamics cause disparity between field observations and model predictions – how well is canopy dynamics simulated by SPA? How is the LAI forced over the canopy layers in SPA? Please elaborate on these aspects in the manuscript and explain how your study improve these shortcomings."

See insertion for performance of SPA.

On the subject of how the presented study improves current shortcomings in LAI modelling:
The authors outline a need for model development around structural and trait responses to drought stress. An exploration of the model structures which would more accurately simulate LAI was outside the scope of the manuscript, but is the subject of ongoing research by the authors.
In addition see other AR9 responses.

plant components, and if the leaf NPP requirement exceeded total NPP for the given timestep, the non-structural C pool was drawn upon (where total NPP was calculated as the difference between simulated GPP and autotrophic respiration) (see supplementary material). The NSC pool serves functions

220    additional to the seasonal redistribution of C (e.g. phloem transport and osmoregulation; Dietze et al., 2014). As such, we assume the NSC pool is stable over time. If the NSC pool becomes depleted, a fraction of NPP is redirected towards NSC storage. Allocation towards NSC storage is executed in subsequent time steps when leaf NPP does not exceed total NPP. Root and wood NPP were calculated from the NPP remaining after leaf allocation. Leaf maintenance respiration was calculated as a function

225    of leaf N content (Reich et al., 2008) and total leaf C stock (see supplementary material). Within SPA, wood and fine root maintenance respiration were simulated as a function of component C stock and a plot specific respiration coefficient. Growth respiration was calculated as fixed fraction of net primary productivity (NPP; (0.28) (Waring and Schlesinger, 1985). Model inputs and outputs are listed in Table 2.

230    *2.2 Model Calibration*

Following data collation to parameterise SPA, the model was calibrated and validated for each plot prior to conducting model experiments. Measurements used to parameterise SPA include: soil texture, soil C stock, leaf N content, LMA, photosynthetic capacity, the fraction of NPP allocated to fine roots and wood, root depth, and foliar, wood and fine root C stocks (Table 2). Soil, wood and fine root C

235    stocks (single point measurements, not timeseries) were initial model inputs and allowed to vary thereafter dependent on simulated C dynamics. Plot specific field measurements of leaf N content are presented in Fyllas et al. (2009), or where absent were retrieved from trait databases using plot species composition (Kattge et al., 2011; Poorter and Bongers, 2006). Photosynthetic capacity estimates ($V_{cmax}$ and $J_{max}$) were derived from leaf N content (Walker et al., 2014), or field measurements (Caxiuanã

240    only). Wood and root respiration measurements were used together with component C stocks to estimate plot specific wood and root respiration coefficients.

The model was driven using hourly meteorological data, retrieved from local weather stations. The number of missing hourly field meteorological measurements across the timeseries varied from 2-40%

**Commented [FS22]:** R1 AR11. In response to the reviewer comment "Line 239 This sentence is not clear."
Edited and expanded to detail additional functions of the NSC pool such as phloem transport and osmoregulation.

**Commented [FS23]:** R1 AR5. In response to the reviewer comment "Several times you briefly mention the C allocation (line 232 to 241, Fig. 2 and Supplement material). In the text you state that allocation to NPPleaf occurs first. Normally NPP is considered a flux, and normally you would allocate to a pool. Thus, do you mean that allocation to the foliar stock occurs first? If assimilation does not provide the C need for allocation to support the LAI, you take from the labile/non-structural carbon pool. However, in the supplement material in the last three equations, you state that if the labile pool has been depleted you allocate from the total NPP. Surely this must only be the case when you have enough NPP to sustain the foliar stock as required by the LAI. Please clarify this in the manuscript."
The reviewer is correct in their summation of the C allocation scheme. Allocation towards NSC storage is executed in subsequent time steps when the NPPleaf requirement does not exceed total NPP. This is now clarified in the manuscript.
Also see supplementary material.

**Commented [FS24]:** R1 AR1. In response to the reviewer comment "The authors investigate the importance of different drivers (LAI, leaf traits, climate) for GPP at both temporal and spatial scale across a drought stress gradient in the amazonian region using the Soil-Plant-Atmosphere (SPA) ecosystem model. The SPA model is applied at 7 sites, using sites specific parameters and is forced with LAI observed from hemispherical photographs. Simulation experiments and machine learning techniques are used to investigate their scientific questions. They find that indirect effects via plant traits and ecosystem structural changes, here expressed as LAI, are found to be the main driver of GPP across a spatial drought gradient, but the sensitivity of GPP to changes in these drivers varied with the gradient. On a sub-annual timescale climatic drivers were found to be more important for GPP. The authors discuss how these direct physiological and indirect mechanism affect GPP but fail to explain the added value of forcing their model with observed LAI and to explain in detail how this forced LAI propagates down the modelling structure of SPA. The manuscript is well written and well structured, however, with many repetitions that should be deleted to make space for more details on your methods. As explained in detail in the comments below, I would like the authors to consider my questions and comments, before I recommend the publication of this manuscript."
AR 1. The brief summary of how LAI is forced within the model (lines 232-237) has been expanded to provide a more in-depth description, and instead included in the '2.1 The Soil Plant Atmosphere model (SPA)' subsection (also see comments AR 5, 9 and 10). In addition, as suggested, we outline the value of using LAI data to force the model.

**Commented [FS25]:** R1 AR35. In response to reviewer comment "Line 195 Please Abbreviate NPP here"

across sites, whilst the frequency of gaps varied from 2-99 yr$^{-1}$. Gaps less than 6 hours in length accounted for between 20-100% of total gaps across plots. Short gaps in air temperature, wind speed, shortwave radiation and vapour pressure deficit measurements (<6 hours), were filled by spline interpolation between existing data. Where local meteorological data was unavailable for a longer period of time, or for gaps in precipitation measurements, hourly spline-interpolated ERA-Interim data were used (Dee et al., 2011). The interpolation of solar radiation estimates accounted for the solar zenith angle. MCWD was calculated for the years 2009-2010, and as the minimum monthly water deficit reached within the year, where monthly water deficit is equal to the previous month's water deficit, plus precipitation, minus evapotranspiration (Aragao et al., 2007). Calculated MCWD was consistent with previously published estimates for all plots excluding Caxiuanã, which were calculated across different years (Malhi et al., (2015), Caxiuanã -203mm, Tambopata -259mm, Kenia -386mm, Tanguro -482mm; this study, Caxiuanã -85±65mm, Tambopata -265±59mm, Kenia 342±146mm, Tanguro 451±73mm).

**Commented [FS26]:** R2 AR10. In response to the reviewer comment "- Line 208. How frequent were data gaps?" Statistics on the frequency of data gaps has now been added to the manuscript.

The simulation of soil water drainage in SPA was calibrated against timeseries of field measurements of soil moisture. Initial investigations comparing modelled soil moisture to monthly field data highlighted an overestimation by SPA. Pre-calibration, SPA soil moisture estimates were on average 11-68% higher than field measurements across plots. The difference between model and field soil moisture estimates increased significantly with MCWD ($R^2$=0.69, p=0.04). The empirical model used in SPA to relate soil texture to water retention (Saxton et al., 1986, eqn. 10) was then calibrated by adjusting the slope of the interaction to better represent soil moisture across tropical soils (to within standard error estimates of mean annual soil moisture).

**Commented [FS27]:** R2 AR11. In response to the reviewers comment "- Line 222. . . .overestimation. . . please quantify."

Leaf litterfall parameters (day of peak leaf fall, leaf fall period and leaf lifespan) were calibrated against field data to accurately simulate litterfall period and amplitude (within standard error estimates of annual litterfall). Wood and fine root biomass turnover rates were estimated assumed proportional ing each forest ecosystem was at steady state to NPP, given the maturity of stands and their disturbance history:

$$turnover\ rate_i \propto \frac{NPP_i}{C\ stock_i}$$

270 Where *i* is wood or fine roots.

Local, monthly LAI estimates derived from hemispherical photographs were scaled to daily estimates via linear interpolation, and used to force simulated LAI. The vertical distribution of leaf area is kept constant, as current field data is insufficient to provide an accurate depiction of how vertical distributions change with canopy density across the MCWD gradient.

275 ~~Leaf NPP was calculated as the difference between the foliar C stock of the previous timestep and that which would equate to field measured LAI. Leaf NPP was allocated prior to other plant components, and if the leaf NPP requirement exceeded total NPP for the given timestep, the non-structural C pool was drawn upon (where total NPP was calculated as the difference between simulated GPP and autotrophic respiration) (see supplementary material). Under the assumption that allocation to NSC is an active process and that the pool serves functions additional to the seasonal redistribution of C (Dietze~~
280  We calculate model uncertainty as a result of input parameters. SPA was forced with the observed LAI timeseries plus and minus the standard error for each plot. Model uncertainty estimates
285 were limited to that derived from LAI as the availability of uncertainty estimates for leaf traits, root depth and root biomass were variable and plot dependent, and there were no uncertainty estimates for hourly meteorological data or soil properties. Model structural uncertainty was not calculated and we recognise that the model error estimates presented are therefore underestimated. With respect to model structural uncertainty, we highlight that the stomatal conductance algorithm embedded within SPA is
290 consistent with leaf and canopy scale observations, and surpasses the performance of the Ball-Berry model where soils experience moisture-stress (Bonan et al., 2014). However, model (and empirical) uncertainty remains around the role of non-structural carbon in regulating water-transport in large trees during drought periods (O'Brien et al., 2014) Furthermore, SPA does not account for hydraulic lift and redistribution of water through the soil profile, which is known to impact water fluxes across the soil-
295 plant-atmosphere continuum in Amazon trees (Oliveira et al, 2005; Wang et al., 2011).

**Commented [FS28]:** R1 AR10. In response to the reviewers comment "NPPLeaf was calculated as the difference between the foliar C stock of the previous time step and that which would equate to field measured LAI. The field measured LAI has a monthly resolution. In principle you would have foliar C stock that could change at every model time step. But if the foliar C stock already equates to the field measured LAI, because the resolution of the forced LAI is monthly, NPPLeaf would just be zero."
This is now clarified in the manuscript.

**Commented [FS29]:** R1 AR9. In response to the reviewer comment "Line 232-234 You state that the mapping of canopy dynamics is critical, and that changes in canopy dynamics cause disparity between field observations and model predictions – how well is canopy dynamics simulated by SPA? How is the LAI forced over the canopy layers in SPA? Please elaborate on these aspects in the manuscript and explain how your study improve these shortcomings."
In addition see other AR9 responses.

**Commented [FS30]:** R1 AR12 & AR16. In response to reviewer comments "Line 243 How was the SPA model calibration constraint by an upper and lower sample error of LAI? Please add clarification to the text."

The model was forced using the observed LAI timeseries plus and minus the standard error for each plot. This is now clarified in the text.

"Line 326 Please explain why the GPPSPA variance is calculated under the LAI standard error."

LAI upper and lower standard error were used to calculate an estimate of SPA uncertainty based on model input data. With regards to other model inputs, the availability of trait uncertainty estimates was variable and plot dependent, and there were no uncertainty estimates for hourly meteorological data. We were consequently limited to LAI estimates.

**Commented [FS31]:** R1 AR23. In response to the reviewers comment "Line 517 In this section, or possible add another section to the discussion, you will also have to address the uncertainties from the intrinsic model behaviour. You recognised already in line 245 that the model uncertainty is underestimated due to the lack of intrinsic model error. However, during your discussion this is not assessed at all. Please assess these uncertainties in particular in relation to the moisture stress and how the plant hydraulics is modelled in SPA and acknowledge its limitations."

**2.3 Model Validation**

[revised manuscript text omitted]

**Commented [FS33]:** R1 AR13. N response to reviewer comment "Line 301-302 These lines are repetitions, and not needed. Please consider deleting."
Deleted as suggested

**Commented [FS34]:** R1 AR36. In response to reviewer comment "Line 307 Add a space between the two sentences."

**Commented [FS35]:** R1 AR14. In response to reviewer comment "Line 308 There is something wrong with the structure of this sentence." Reworked as suggested

**Commented [FS36]:** R1 AR31. In response to reviewer comment "In general, Please reconsider the usage of the word whilst – it reads a little pretentious."
Adjustments throughout

[revised manuscript text omitted]

**Commented [FS38]:** R1 AR31. In response to the reviewers comment "In general, Please reconsider the usage of the word whilst – it reads a little pretentious." Adjustments throughout

**Commented [FS39]:** R2 AR13. In response to the reviewers comment "- Line 359. Please clarify how this can be seen in table 3."
This was an error and should have referenced the preceding section ('3.4 LAI and Leaf Trait Trends along the MCWD gradient'). Now corrected.

). The sensitivity of GPP to LAI exhibited a positive, non-significant correlation with MCWD ($R^2$=0.88, p=0.06). Reflecting LAI trends, the sensitivity of GPP to differences in photosynthetic capacity (per unit μmol C g⁻¹) was similarly highest at Caxiuanã , and decreased linearly (though not significantly) across the MCWD gradient ($R^2$=0.83, p=0.09). Tambopata plots, which have high LAI but shallow rooting depth, were most sensitive to differences in climate (per unit MCWD mm) . Kenia plots, which have similarly shallow rooting depth but low LAI, were the least sensitive . The sensitivity of GPP to differences in rooting depth (per m rooting depth) was highest at Tanguro and Tambopata , and lowest at Caxiuanã and Kenia .

**3.5.3 Experiment 3. Drivers of Sub-Annual Variation in GPP**

In contrast to drivers of spatial variation in GPP, on a sub-annual timescale LAI had less explanatory power than climate (Tables 6 and 7). The relative importance of solar radiation in driving monthly GPP increased significantly with MCWD ($R^2$ = 0.90, p=<0.001),  as the relative importance of LAI declined ($R^2$=0.72, p=0.015). The relative importance of VPD did not vary directionally across the MCWD gradient ($R^2$=0.10, p=0.49). Both precipitation and air temperature had little effect on monthly GPP, though it is noted that a significant interaction existed between both precipitation and VPD (p<0.001) and air temperature and shortwave radiation (p<0.001). Furthermore, temperature varied least across plots in comparison to other climate forcings (standard deviation as a percentage of the mean; temperature 9.8%, VPD 73%, precipitation 192%, shortwave radiation 34%). As such, seasonal changes in the relative importance of temperature and precipitation were not investigated further. The relative importance of LAI, VPD and solar radiation shifted seasonally, reflecting changes in the availability of light and water. Solar radiation was typically the most important driver of monthly GPP during the wet

**Commented [FS40]:** R1 AR31. In response to reviewer comment "In general, Please reconsider the usage of the word whilst – it reads a little pretentious."
Adjustments throughout

**Commented [FS41]:** R2 AR14. In response to the reviewers comment "- Section 3.5.2. This section is difficult to read because of the many numbers. Is presenting them in a table of figure an option?"
The authors have moved the site specific Δ values to Figure 6.

**Commented [FS42]:** R1 AR31. In response to reviewer comment "In general, Please reconsider the usage of the word whilst – it reads a little pretentious."
Adjustments throughout

[revised manuscript text omitted]

495 Evidence of changes in LAI in response to precipitation regime has been presented across multiple ecosystems and over time (Grier and Running, 1977, Schleppi et al., 2011, Iio et al., 2014, Dobbertin et al., 2010, Wright et al., 2013). Amazonian forest throughfall exclusion experiments identified a decline in LAI with the onset of reduced soil water (Fisher et al., 2007, Meir et al., 2008, Brando et al., 2008). At Caxiuanã, over a 4-year period, observed leaf area was 20-30% lower than the control stand 500 (Meir et al., 2009), with long-term reductions estimated at between 10-15% (Rowland et al., 2015a). Investigations show that declines in LAI are not caused by increased leaf turnover due to drought stress, but instead are the result of lower leaf production (Nepstad et al., 2002, Schuldt et al., 2011), suggesting an active response of plant allocation strategy to water availability. Concurrently, after 15 years under throughfall exclusion, Rowland et al. (2018) found that leaf litterfall still remained consistently lower. 505 Reported trends in canopy dynamics are therefore in accordance with our findings, and indicate that LAI is a key response mechanism to precipitation regime. Other studies such as da Costa et al.

**Commented [FS44]:** R1 AR31. In response to reviewer comment "In general, Please reconsider the usage of the word whilst – it reads a little pretentious."
Adjustments throughout

**Commented [FS45]:** R1 AR31. In response to reviewer comment "In general, Please reconsider the usage of the word whilst – it reads a little pretentious."
Adjustments throughout

**Commented [FS46]:** R1 AR18. In response to the reviewer comment

"Line 437-449 You mention how changes in LAI is a response to precipitations regimes, and even call it a key response mechanism. Then, one could infer that it is just as much changes in precipitation that explains the changes in GPP as it is LAI. You state that changes in LAI is an indirect effect from climate — although the impact might be somewhat delayed, but did you investigate lagged correlations and variances between precipitation and GPP?"

With respect to annual timescales the field data used were across 2 years only, limiting the scope to test for lag effects. However, this could certainly prove an interesting investigation for the future. Across sub-annual timescales we are unable to compare field estimates of GPP (due to the nature of biometric estimates), however model experiment 3 found that the effect of moisture stress was better captured via VPD (line 392).

(2018) have similarly pointed towards structural responses as the principal determinant of variation in GPP,  they identify changes in sapwood area as the main driver, rather than LAI. We suggest that whilst sapwood area may be more important in shaping the response to  short term changes in precipitation,  over longer timescales emergent canopy properties (LAI) drive GPP trends.

Photosynthetic capacity also proved an important driver of spatial variation in GPP across the MCWD gradient. Our results are consistent with a number of Amazon-based studies linking leaf traits to productivity (Aragao et al., 2009, Cleveland et al., 2011, Castanho et al., 2013). Interestingly, the observed shifts in photosynthetic capacity along the gradient had a compensatory effect on the GPP-MCWD interaction. Reductions in GPP under high drought stress were alleviated by higher photosynthetic capacitance. Similarly, shifts in photosynthetic capacity in response to temperature have been reported to reduce spatial variation in GPP across a tropical elevation gradient (Bahar et al., 2017, van de Weg et al., 2013). Consistent with Fyllas et al. (2017), our results also show that the effect of climatic forcings on carbon fluxes can be successfully captured through spatial variation in canopy dynamics and leaf traits. However, as we have focused on the role of leaf traits in the absence of carbon cycle feedbacks, we do not take into account the effect of concurrent shifts in LMA and leaf lifespan, which together influence canopy carbon economics (Wright et al., 2004, Osnas et al., 2013, McMurtrie and Dewar, 2011).

Root depth, root biomass and soil properties had little direct effect on spatial variation in GPP. we recognise that the difficulty in measuring root depth and biomass (Metcalfe et al., 2007) adds uncertainty to our results, however, the findings do not reflect the importance of belowground functioning highlighted by other studies (Fisher et al., 2007, Metcalfe et al., 2008, Baker et al., 2008, Phillips et al., 2009, Ichii et al., 2007). Notably, a number of GEM plots had hard pan layers (Quesada et al., 2012) so they may be acclimated to operate in shallow rooting zones, and are therefore not

Commented [FS47]: R1 AR31. In response to reviewer comment "In general, Please reconsider the usage of the word whilst – it reads a little pretentious." Adjustments throughout

Commented [FS48]: R1 AR19. In response to the reviewer comment "Line 452 You do not have a forest at steady state, if it is changing and experiencing trends. Please explain what you mean by a steady state forest – is it continuous cover, constant number of trees etc.? Or better yet refrain from using the term." Term omitted

Commented [FS49]: Moved to limitations section.

Commented [FS50]: R1 AR31. In response to reviewer comment "In general, Please reconsider the usage of the word whilst – it reads a little pretentious." Adjustments throughout

Commented [FS51]: R1 AR37. In response to reviewer comment "Line 473-774 Add they to the sentence: . . . so they may be acclimated. . ."

[revised manuscript text omitted]

**Commented [FS53]:** R1 AR22 In response to the reviewer comment "Line 515 What do you mean by GPP is demand limited across spatial scale? Is it the atmospheric transpirational demand? And how does this relate to your already conducted analysis of you results? According to your own analysis, LAI is explaining most of the variation across the spatial scale (sec. 3.5.1, sec. 3.5.2, and 4.2). You must elaborate on the statement."
This statement referred to the role of ecosystem structure and traits (i.e. demand) in determining spatial variation in GPP, in comparison to the supply limit observed across sub-annual timescales. However we recognise that the statement was unclear and have amended it.

**Commented [FS54]:** R1 AR20. In response to reviewer comment "Line 465-468 With the SPA model you are not able to quantify the impact of soil nutrient on the GPP-MCWD interactions because you lack nutrient cycling in the SPA model. However, according to Table 1 there is a huge P gradient across your sites, therefore please discuss how this could possible affect your results."
Discussion on nutrient limitation is now included.

**Commented [FS55]:** Moved from section 4.4

in scaling biometric measurements to plot level (e.g. uncertainty in using estimated total woody surface area to scale stem $CO_2$ efflux measurements).

Given the importance of LAI in driving variation in GPP, data on canopy dynamics is critical to constrain carbon flux estimates across the Amazon basin. Our approach utilised field estimates of LAI from hemispherical photographs to constrain model simulations. The accuracy and spatial validity of indirect estimates of LAI has been questioned at higher leaf areas (Bréda, 2003, Jonckheere et al., 2004, Weiss et al., 2004). In this study, we expect that if field measurements of LAI were underestimated at higher leaf areas, the proportion of spatial variation in GPP explained by LAI would increase, as a result of increased variation in both field-measured and model simulated GPP. Yet, our highest estimates of LAI (Caxiuanã 5.11 ± 1.41 $m^2m^{-2}$) align with destructive sampling measurements from a terra-firme Amazon forest (McWilliam et al. (1993) 5.7 ± 0.5 $m^2m^{-2}$). Furthermore, a comparison of LAI estimation approaches (Asner et al., 2003) suggested that indirect methods were appropriate for broadleaved forests, and presented no statistical difference between destructive harvesting and indirect methods. However, the use of ground measurements is limited to smaller spatial scales, and LAI estimates across the basin are needed to constrain carbon flux estimates.  Though the interpretation of forest responses to drought stress through remote sensing approaches have caused controversy (Asner and Alencar, 2010, Saleska et al., 2007, Samanta et al., 2010), an increase in canopy mapping through satellite missions could be instrumental to efforts aiming to better understand LAI dynamics. Current and upcoming satellite missions including FLEX (FLuorescence EXplorer), GEDI (Global Ecosystem Dynamics Investigation) and Sentinel will offer opportunity for new insights into changes in leaves *in-situ*, vertical canopy structure, and temporal variability via repeat measurements (Morton, 2016, Drusch et al., 2017, Pettorelli et al., 2018). Efforts to map trait distributions will also prove important (Kattge et al., 2011, Asner et al., 2015) given their role in driving variation in GPP.

**5. Conclusion**

We show that indirect effects of climate (via ecosystem structure and trait responses) exceed direct effects (via physiological responses) in driving spatial variation in GPP across an Amazon MCWD gradient (Q1). Conversely, across sub-annual timescales, the reverse was true (Q3). The relative

**Commented [FS56]:** R1 AR15. In response to the reviewers comment "Line 320-325 The correlation between GEM estimated and SPA simulation GPP are non-significant and moderate. As GPP is imperative for your analysis, have you considered the impact it might have on you results? Have you investigated how the GEM estimated GPP relates to the LAI form the hemispherical photographs?" Discussion on the GEM-SPA GPP comparison and its potential impact is now included in the limitation sections. In response to the reviewers question, GPP estimates are not significantly correlated with LAI ($R^2$=0.18, p=0.33), likely due to concurrent shift towards higher photosynthetic capacity at drier sites.

**Commented [FS57]:** R1 AR31. In response to reviewer comment "In general, Please reconsider the usage of the word whilst – it reads a little pretentious." Adjustments throughout

sensitivity of GPP to changes in direct and indirect forcings shifted across the MCWD gradient and was

640 dependent on water availability, demand and acquisition potential (Q2). We, identifying the potential vulnerability of forests with a high evaporative potential (i.e. high LAI), but low water uptake capacity (i.e. shallow rooting depth), to changes in precipitation regime. Given the role of LAI in driving GPP across the drought stress gradient, we highlight a requisite for improved mapping of canopy dynamics (via remote sensing). We propose that ecosystem model development should focus on the integration

645 of structural and trait responses to drought stress (alongside physiological responses). The inclusion of both direct and indirect effects of climate in ecosystem models, would reduce current uncertainty in predicted annual and sub-annual GPP for tropical forests.

650

**Supplementary Material**

Supplementary material is included in a separate document.

**Authorship Contributions** Sophie Flack-Prain, Mathew Williams and Patrick Meir conceived the research questions. Data used in model calibration and validation was collected by Yadvinder Malhi

655 and associates. Model experiments were designed and conducted by Sophie Flack-Prain with contributions from Mathew Williams and Thomas L. Smallman. Sophie Flack-Prain and Mathew Williams prepared the manuscript with active contributions from all co-authors.

**Conflict of Interest Statement:** There are no conflicts of interest to disclose.

**Acknowledgements:** The authors would like to thank the Global Ecosystems Monitoring network team

660 for the field data used in this study, collected through funding from NERC and the Gordon and Betty Moore Foundation, and an ERC Advanced Investigator Award to YM (GEM-TRAIT). The authors would also like to thank the PhD project funding body, the UK Natural Environment Research Council

Commented [FS58]: R1 AR24 In response to the reviewer comment "Line 544-546 Something is wrong with this part of sentences. Please rephrase."
Rephrased.

[revised manuscript text omitted]

**Commented [FS62]:** R1 AR25. In response to the reviewers comment "Line 576 According to the manuscript preparation guidelines for authors for Biogeoscience, Journal names are to be abbreviated according to the Journal Title Abbreviations by Caltech Library (see https://www.biogeosciences.net/for_authors/manuscript_preparation.html )."

Now corrected

**Tables**

Table 1. Amazonian Forest Inventory Network (RAINFOR) site code and eEnvironmental characteristics of GEM network Amazon permanent sample plots across the MCWD gradient. Meteorological data is from local weather stations, gap filled with ERA interim data for the years 2009-2010 (Dee et al., 2011).

| Plot name | Caxiuanã Control | Caxiuanã Tower | Tambopata V | Tambopata VI | Kenia Wet | Kenia Dry | Tanguro Control |
|---|---|---|---|---|---|---|---|
| RAINFOR-site code | CAX04 | CAX06 | TAM05 | TAM06 | KEN01 | KEN02 | --- |
| Latitude | -1.716 | -1.737 | -12.831 | -12.839 | -16.016 | -16.016 | -13.077 |
| Longitude | -51.457 | -51.462 | -69.271 | -69.296 | -62.73 | -62.73 | -52.386 |
| Elevation (m.a.s.l) | 47 | | 223 | | 384 | | 385 |
| Mean Maximum Climatological Water Deficit (mm) | -85.5 | | -256 | | -342 | | -498 |

Commented [FS63]: R1 AR46. In response to reviewer comment "Would it be possible to add species composition or just dominant species at each site to the table?" This could be added. However, species richness varies between 65 and 195 across plots, with the most common species typically accounting for less than 20% of stems.

Commented [FS64]: R1 AR42. In response to reviewer comment "Line 996-999 Table 1: Please include the abbreviation of RAINFOR in the Table 1 text."

Commented [FS66]: R1 AR44. In response to reviewer comment "Is there no site code for Tanguro?" No, it is not on the RAINFOR database (though RAINFOR does hold data on other Tanguro plots).

Commented [FS65]: R1 AR43. In response to reviewer comment "Should RAINFOR be above the second horizontal line?" No, the row name should read 'RAINFOR site code'. This has be resolved by adding a hyphen.

[revised manuscript text omitted]

**Commented [FS71]:** R1 AR48. In response to reviewer comment "Line 1112 Please correct nodule to module." This was supposed to refer to 'nodule' however as this does not read well we have changed it to node.

**Commented [FS72]:** R1 AR31. In response to reviewer comment "In general, Please reconsider the usage of the word whilst – it reads a little pretentious." Adjustments throughout

[revised manuscript text omitted]

**SPA Non-Structural Carbon Pool**

For the purpose of the presented study, LAI was forced in model runs. Where leaf NPP requirements could not be met by daily C assimilated, leaf growth was supplemented by the labile carbon pool as follows:

$$NPP_i = GPP_i - Ra_i \qquad (5)$$

$$NPP_{leaf\,i} = (LAI_i - LAI_{i-1}) \times LCA \qquad (6)$$

$$If \; NPP_{leaf\,i} > NPP_i \Rightarrow \qquad (7)$$

**Commented [FS75]:** R1 AR29. In response to reviewer comment "Please consider numerating your equations If the LAI is forced using monthly time series, then how does the LAI change between the daily time steps in the calculations for NPPLeaf i? Is it because you nudge the LAI, and hence not force it at every time step? Please clarify."
Numbering added. See AR 10 for clarification on LAI change at each timestep.

**Commented [FS76]:** R1 AR30. In response to reviewer comment "As mentioned already, the three latter equations are confusing. If the labile pool is depleted (NSCi < NSCi−1) then you allocate from the total NPP pool to the labile pool. I assume that this is only the case when NPPLeafi is met by the daily assimilation? Please clarify and complete your sets of equations for all cases"
Please see AR 5.

**Commented [FS77]:** R1 AR28 Corrected

$$NSC_i = NSC_{i-1} - (NPP_{leaf\,i} - NPP_i) \quad (8)$$

$$\text{If } NSC_i < NSC_{cap} \text{ AND } NPP_{leaf\,i} < NPP_i) \Rightarrow \quad (9)$$

$$NPP_{i+1} = NPP_{i+1} - NSC_{frac} \quad (10)$$

$$NSC_{i+1} = NSC_i + NSC_{frac} \quad (11)$$

Where:

NPP is modelled net primary productivity, GPP is modelled gross primary productivity, Ra is modelled autotrophic respiration, $NPP_{leaf}$ is modelled leaf net primary productivity, LAI is the field estimated leaf area index, LCA is the field estimated leaf C content per unit leaf area, NSC is the non-structural carbon pool, $NSC_{frac}$ is the fraction of NPP redirected towards the NSC pool, $NSC_{cap}$ is the NSC pool capacitance, and i is the daily timestep.

**Calculation of Maximum Climatological Water Deficit**

We calculate maximum climatological water deficit (MCWD) in line with the equations presented in Aragao et al. (2007):

$$\text{If } WD_{n-1} - E + P_n < 0 \Rightarrow \quad (12)$$

$$WD_n = WD_{n-1} - E + P_n \quad (13)$$

$$\text{Else } WD_n = 0 \quad (14)$$

Where:

WD is the water deficit for each month (n), E is evapotranspiration (assumed to be 100 mm month$^{-1}$), and P is precipitation (mm month$^{-1}$).

**Commented [FS78]:** R1 AR5. In response to the reviewer comment "Several times you briefly mention the C allocation (line 232 to 241, Fig. 2 and Supplement material). In the text you state that allocation to NPPleaf occurs first. Normally NPP is considered a flux, and normally you would allocate to a pool. Thus, do you mean that allocation to the foliar stock occurs first? If assimilation does not provide the C need for allocation to support the LAI, you take from the labile/non-structural carbon pool. However, in the supplement material in the last three equations, you state that if the labile pool has been depleted you allocate from the total NPP. Surely this must only be the case when you have enough NPP to sustain the foliar stock as required by the LAI. Please clarify this in the manuscript."
The reviewer is correct in their summation of the C allocation scheme. Allocation towards NSC storage is executed in subsequent time steps when the NPPleaf requirement does not exceed total NPP. This is now clarified in these equations and in the manuscript.

**Experimental Model Runs**

**Climate Impact**

| Climate | Soil Properties | LAI | Root Biomass | Rooting Depth | Photosynthetic Capacity |
|---|---|---|---|---|---|
| CAX04 | $Plot_n$ | $Plot_n$ | $Plot_n$ | $Plot_n$ | $Plot_n$ |
| CAX06 | $Plot_n$ | $Plot_n$ | $Plot_n$ | $Plot_n$ | $Plot_n$ |
| TAM05 | $Plot_n$ | $Plot_n$ | $Plot_n$ | $Plot_n$ | $Plot_n$ |
| TAM06 | $Plot_n$ | $Plot_n$ | $Plot_n$ | $Plot_n$ | $Plot_n$ |
| KEN01 | $Plot_n$ | $Plot_n$ | $Plot_n$ | $Plot_n$ | $Plot_n$ |
| KEN02 | $Plot_n$ | $Plot_n$ | $Plot_n$ | $Plot_n$ | $Plot_n$ |
| Tanguro | $Plot_n$ | $Plot_n$ | $Plot_n$ | $Plot_n$ | $Plot_n$ |

**Soil Properties Impact**

| Climate | Soil Properties | LAI | Root Biomass | Rooting Depth | Photosynthetic Capacity |
|---|---|---|---|---|---|
| $Plot_n$ | CAX04 | $Plot_n$ | $Plot_n$ | $Plot_n$ | $Plot_n$ |
| $Plot_n$ | CAX06 | $Plot_n$ | $Plot_n$ | $Plot_n$ | $Plot_n$ |
| $Plot_n$ | TAM05 | $Plot_n$ | $Plot_n$ | $Plot_n$ | $Plot_n$ |
| $Plot_n$ | TAM06 | $Plot_n$ | $Plot_n$ | $Plot_n$ | $Plot_n$ |
| $Plot_n$ | KEN01 | $Plot_n$ | $Plot_n$ | $Plot_n$ | $Plot_n$ |
| $Plot_n$ | KEN02 | $Plot_n$ | $Plot_n$ | $Plot_n$ | $Plot_n$ |
| $Plot_n$ | Tanguro | $Plot_n$ | $Plot_n$ | $Plot_n$ | $Plot_n$ |

**LAI Impact**

| Climate | Soil Properties | LAI | Root Biomass | Rooting Depth | Photosynthetic Capacity |
|---|---|---|---|---|---|
| $Plot_n$ | $Plot_n$ | CAX04 | $Plot_n$ | $Plot_n$ | $Plot_n$ |
| $Plot_n$ | $Plot_n$ | CAX06 | $Plot_n$ | $Plot_n$ | $Plot_n$ |
| $Plot_n$ | $Plot_n$ | TAM05 | $Plot_n$ | $Plot_n$ | $Plot_n$ |
| $Plot_n$ | $Plot_n$ | TAM06 | $Plot_n$ | $Plot_n$ | $Plot_n$ |
| $Plot_n$ | $Plot_n$ | KEN01 | $Plot_n$ | $Plot_n$ | $Plot_n$ |
| $Plot_n$ | $Plot_n$ | KEN02 | $Plot_n$ | $Plot_n$ | $Plot_n$ |
| $Plot_n$ | $Plot_n$ | Tanguro | $Plot_n$ | $Plot_n$ | $Plot_n$ |

**Root Biomass Impact**

| Climate | Soil Properties | LAI | Root Biomass | Rooting Depth | Photosynthetic Capacity |
|---|---|---|---|---|---|
| $Plot_n$ | $Plot_n$ | $Plot_n$ | CAX04 | $Plot_n$ | $Plot_n$ |
| $Plot_n$ | $Plot_n$ | $Plot_n$ | CAX06 | $Plot_n$ | $Plot_n$ |
| $Plot_n$ | $Plot_n$ | $Plot_n$ | TAM05 | $Plot_n$ | $Plot_n$ |
| $Plot_n$ | $Plot_n$ | $Plot_n$ | TAM06 | $Plot_n$ | $Plot_n$ |
| $Plot_n$ | $Plot_n$ | $Plot_n$ | KEN01 | $Plot_n$ | $Plot_n$ |
| $Plot_n$ | $Plot_n$ | $Plot_n$ | KEN02 | $Plot_n$ | $Plot_n$ |
| $Plot_n$ | $Plot_n$ | $Plot_n$ | Tanguro | $Plot_n$ | $Plot_n$ |

**Rooting Depth Impact**

| Climate | Soil Properties | LAI | Root Biomass | Rooting Depth | Photosynthetic Capacity |
|---|---|---|---|---|---|
| $Plot_n$ | $Plot_n$ | $Plot_n$ | $Plot_n$ | CAX04 | $Plot_n$ |
| $Plot_n$ | $Plot_n$ | $Plot_n$ | $Plot_n$ | CAX06 | $Plot_n$ |
| $Plot_n$ | $Plot_n$ | $Plot_n$ | $Plot_n$ | TAM05 | $Plot_n$ |
| $Plot_n$ | $Plot_n$ | $Plot_n$ | $Plot_n$ | TAM06 | $Plot_n$ |
| $Plot_n$ | $Plot_n$ | $Plot_n$ | $Plot_n$ | KEN01 | $Plot_n$ |
| $Plot_n$ | $Plot_n$ | $Plot_n$ | $Plot_n$ | KEN02 | $Plot_n$ |
| $Plot_n$ | $Plot_n$ | $Plot_n$ | $Plot_n$ | Tanguro | $Plot_n$ |

**Photosynthetic Capacity Impact**

| Climate | Soil Properties | LAI | Root Biomass | Rooting Depth | Photosynthetic Capacity |
|---|---|---|---|---|---|
| $Plot_n$ | $Plot_n$ | $Plot_n$ | $Plot_n$ | $Plot_n$ | CAX04 |
| $Plot_n$ | $Plot_n$ | $Plot_n$ | $Plot_n$ | $Plot_n$ | CAX06 |
| $Plot_n$ | $Plot_n$ | $Plot_n$ | $Plot_n$ | $Plot_n$ | TAM05 |
| $Plot_n$ | $Plot_n$ | $Plot_n$ | $Plot_n$ | $Plot_n$ | TAM06 |
| $Plot_n$ | $Plot_n$ | $Plot_n$ | $Plot_n$ | $Plot_n$ | KEN01 |
| $Plot_n$ | $Plot_n$ | $Plot_n$ | $Plot_n$ | $Plot_n$ | KEN02 |
| $Plot_n$ | $Plot_n$ | $Plot_n$ | $Plot_n$ | $Plot_n$ | Tanguro |

Figure S1. Model experimental design to apportion variation in simulated GPP to that driven by differences in (i) climate, (ii) soil properties, (iii) LAI, (iv) root biomass and (v) rooting depth, and (vi) trait responses driven by photosynthetic capacity ($V_{cmax}$ and $J_{max}$). For a given plot i.e. $Plot_n$ (CAX04, CAX06, TAM05, TAM06, KEN01, KEN02, Tanguro), model inputs (i-vi) were alternated to that of all other plots, and the simulated GPP retrieved.

> **Commented [FS79]:** R2 AR4. In response to the reviewer comment "- The model methodology is explained insufficiently to allow for independent reproducibility of the results and for understanding what the authors really did."
>
> A new figure has been added to the supplementary material detailing the inputs of each model run used in the analysis. We hope this figure helps sufficiently explain our methodology. However, we would be happy to consider further changes if the reviewer could provide more information on which details they believe to be missing.

**References**

[revised manuscript text omitted]